# PUe: Biased Positive-Unlabeled Learning Enhancement by Causal Inference

**Xutao Wang, Hanting Chen, Tianyu Guo, Yunhe Wang**[*]
Huawei Noah's Ark Lab.
{xutao.wang,chenhanting,tianyu.guo,yunhe.wang}@huawei.com,

## Abstract

Positive-Unlabeled (PU) learning aims to achieve high-accuracy binary classification with limited labeled positive examples and numerous unlabeled ones. Existing cost-sensitive-based methods often rely on strong assumptions that examples with an observed positive label were selected entirely at random. In fact, the uneven distribution of labels is prevalent in real-world PU problems, indicating that most actual positive and unlabeled data are subject to selection bias. In this paper, we propose a PU learning enhancement (PUe) algorithm based on causal inference theory, which employs normalized propensity scores and normalized inverse probability weighting (NIPW) techniques to reconstruct the loss function, thus obtaining a consistent, unbiased estimate of the classifier and enhancing the model's performance. Moreover, we investigate and propose a method for estimating propensity scores in deep learning using regularization techniques when the labeling mechanism is unknown. Our experiments on three benchmark datasets demonstrate the proposed PUe algorithm significantly improves the accuracy of classifiers on non-uniform label distribution datasets compared to advanced cost-sensitive PU methods. Codes are available at `https://github.com/huawei-noah/Noah-research/tree/master/PUe` and `https://gitee.com/mindspore/models/tree/master/research/cv/PUe`.

## 1 Introduction

In the era of big data, deep neural networks have achieved outstanding performance across various tasks, even surpassing human performance in many instances, particularly in traditional binary classification problems. The success of these deep neural networks often hinges on supervised learning using large quantities of labeled data. However, in reality, acquiring even binary labels can be challenging. For instance, in recommendation systems, users' multiple clicks on films may be considered as positive samples. Nonetheless, all other films cannot be assumed uninteresting and thus should not be treated as negative examples; instead, they should be regarded as unlabeled ones.

The same issue emerges in text classification, where it is typically more straightforward to define a partial set of positive samples. However, due to the vast diversity of negative samples, it becomes difficult or even impossible to describe a comprehensive set of negative samples that represent all content not included in the positive samples. Similar situations occur in medical diagnostics, malicious URL detection, and spam detection, where only a few labeled positives are available amidst a plethora of unlabeled data. This scenario is a variant of the classical binary classification setup, known as PU. In recent years, there has been a growing interest in this setting. Positive-Unlabeled (PU) learning primarily addresses the challenge of learning binary classifiers solely from positive samples and unlabeled data.

---

[*]Corresponding Author.

37th Conference on Neural Information Processing Systems (NeurIPS 2023).

In previous research, numerous PU algorithms have been developed, with cost-sensitive PU learning emerging as a popular research direction. Methods such as [19, 18, 6] reweight positive and negative risks by hyper-parameters and minimize it. In addition, Self-PU [4] introduces self-supervision to nnPU through auxiliary tasks, including model calibration and distillation; ImbPU [23] oversamples and modifies sample weights to address unbalanced data. Dist-PU[24] corrects the negative preference of the classification model through prior information. PUSB [15] maintains the order-preserving assumption. However, these methods necessitate the assumption that their checked set is uniformly sampled from the population, or else the PU learning risk estimator ceases to be an unbiased or consistent estimator, otherwise resulting in reduced model accuracy.

In reality, the labeled set is often biased and does not conform to the Selected Completely At Random (SCAR) assumption [10, 3], which posits that the observed labeled examples are a random subset of the complete set of positive examples. Consequently, it is essential to relax the assumption of the labeled set and replace it with a more general assumption about the labeling mechanism: the probability of selecting positive examples to be labeled depends on their attribute values, known as the Selected At Random (SAR) assumption [10].

The work most closely related to us is the paper [7] by W Gerych et al. The focus of this article is to develop a method for generating identifiable propensity scores. Moreover, this paper proves that the general propensity score is unidentifiable in any standard PU data assumption except for the probability gap scenario. In this paper, two sets of assumptions are proposed to estimate propensity scores: one is Local Certainty and the other is Probabilistic Gap. Under the local certainty hypothesis, there is no overlap between the positive and negative hypotheses, $p(x^P|y = 0) = 0$. Under the Probabilistic Gap assumption, it is assumed that $e = k * p(y = 1|x)$ is linear and there is an anchor point.

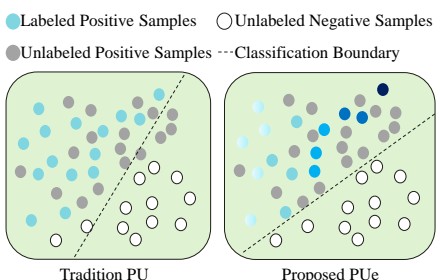

Figure 1: Our method and traditional PU method classification diagram, our solution uses reweighting to make the classification plane more accurate.

However, the above assumptions are too strict. We show in the appendix that the estimated propensity scores cannot be completely unbiased. To address the aforementioned problem, we propose a causal-inference-inspired PU learning framework, termed PUe. Our approach aims to weight the original PU learning risk estimator under biased conditions, thereby obtaining a new unbiased or consistent PUe learning risk estimator, which is achieved through the inverse probability weighting technique of known normalized propensity scores. Given that propensity scores for samples are typically unknown, we apply regularization techniques to the deep learning classifier to estimate the propensity score of each sample in the marker set. An illustration of the proposed method is shown in Figure 1. We also provide an estimate of the generalization error bound for empirical risk minimization. Experimental results demonstrate that the proposed PUe framework can be applied to advanced PU algorithms, such as PUbN and Dist-PU, to enhance their performance in the SAR scenario. Our main contributions are summarized as follows:

•We relax our assumption of local determinism.

•We use a deep learning model to estimate propensity scores instead of a linear model.

•Our propensity score estimation method can be extended to the case of negative classes (e.g. PUbNe).

•Our new algorithm can simply be coupled with most cost-sensitive algorithms to improve performance. The PUe algorithm achieved superior results over existing solutions on standard benchmarks (MNIST, CIFAR10) and the Alzheimer's Disease Neuroimaging Initiative (ADNI) database.

## 2   Methodology

In this section, we conduct a thorough review of existing PU algorithms, examining their limitations and challenges, particularly in the context of biased labeling scenarios that result in diminished accuracy. To address these issues, we introduce a novel PUe algorithm, specifically designed to

overcome the aforementioned limitations, thus enhancing the performance of PU algorithms under biased labeling conditions.

## 2.1 Review of PU classification

In standard PN classification, let $\boldsymbol{x} \in \mathbb{R}^d$ and $y \in \{+1, -1\}$ be the input samples and its corresponding labels. We are given positive data and negative data that are sampled independently from $p_P(\boldsymbol{x}) = p(\boldsymbol{x}|y = +1)$ and $p_N(\boldsymbol{x}) = p(\boldsymbol{x}|y = -1)$ as $\chi_P = \{\boldsymbol{x}_i^P\}_{i=1}^{n_P}$ and $\chi_N = \{\boldsymbol{x}_i^N\}_{i=1}^{n_N}$. We denote the class prior probability as $\pi = p(y = 1)$, where we follow the convention to assume $\pi$ as known throughout the paper [16]. In fact, in the paper [14] by S Jain et al., it is pointed out that class prior can be estimated in the case of the proportion of classes that are biased in the labelling data. Let $g : \mathbb{R}^d \to \mathbb{R}$ be the binary classifier and $\theta$ be its parameter and the $L : \mathbb{R} \times \{+1, -1\} \to \mathbb{R}_+$ be a loss function.

Let us denote by $R_P(g, +1) = \mathbb{E}_{\boldsymbol{x} \sim p_P(\boldsymbol{x})}[L(g(\boldsymbol{x}, +1))]$, $R_N(g, -1) = \mathbb{E}_{\boldsymbol{x} \sim p_N(\boldsymbol{x})}[L(g(\boldsymbol{x}, -1))]$. $R_P(g, +1)$ is the expected risk loss of a positive sample and $R_N(g, -1)$ is the expected risk loss of a negative sample.

The risk of classifier $g$, $\hat{R}_{PN}(g)$ can be approximated by:

$$\hat{R}_{PN}(g) = \pi \hat{R}_P(g, +1) + (1 - \pi)\hat{R}_N(g, -1), \tag{1}$$

where $\hat{R}_P(g, +1) = \frac{1}{n_P} \sum_{i=1}^{n_P} L(g(\boldsymbol{x}_i^P), +1)$ and $\hat{R}_N(g, -1) = \frac{1}{n_N} \sum_{i=1}^{n_N} L(g(\boldsymbol{x}_i^N), -1)$. By minimizing $\hat{R}_{PN}(g)$. We obtain the ordinary empirical risk minimizer $\hat{g}_{PN}$.

In standard PU classification, instead of N data $\chi_N$, we only have a set of U samples drawn from the marginal density $p(\boldsymbol{x})$, $\chi_U = \{\boldsymbol{x}_i^U\}_i^{n_U} \sim p(\boldsymbol{x})$. Because of the fact $p(\boldsymbol{x}) = \pi p_P(\boldsymbol{x}) + (1 - \pi)p_N(\boldsymbol{x})$, we have the unbiased risk estimator, which key idea is to use the following equality:

$$(1 - \pi)R_N(g, -1) = R_U(g, -1) - \pi R_P(g, -1). \tag{2}$$

where $R_U(g, -1) = \mathbb{E}_{x \sim p(\boldsymbol{x})}[L(g(\boldsymbol{x}), -1)]$ and $R_P(g, -1) = \mathbb{E}_{x \sim p_P(\boldsymbol{x})}[L(g(\boldsymbol{x}), -1)]$. As a result, we can approximate the unbiased classification risk (1) by:

$$\hat{R}_{uPU}(g) = \pi \hat{R}_P(g, +1) + \hat{R}_U(g, -1) - \pi \hat{R}_P(g, -1), \tag{3}$$

where $\hat{R}_P(g, -1) = \frac{1}{n_P} \sum_{i=1}^{n_P} L(g(\boldsymbol{x}_i^P), -1)$ and $\hat{R}_U(g, -1) = \frac{1}{n_U} \sum_{i=1}^{n_U} L(g(\boldsymbol{x}_i^U), -1)$. By minimizing $\hat{R}_{PU}(g)$ we obtain the ordinary empirical risk minimizer $\hat{g}_{PU}$.

In theory, the risk $(1 - \pi)R_N(g, -1) = R_U(g, -1) - \pi R_P(g, -1)$ is also positive. However, if the model of $g$ is too flexible, $\hat{R}_{PU}(\hat{g}_{PU})$ indeed goes negative and the model severely overfits the training data. We can use the non-negative risk estimator for PU learning to alleviate overfitting[16]:

## 2.2 PU Classification in Biased Scenarios

PU learning mostly assumes that it runs under an ideal situation, that is, all labeled samples are selected completely at random from all positive samples, which is also called The Selected Completely At Random (SCAR).

**Defintion 1. SCAR**(Selected Completely At Random [6]): Labeled examples are selected completely at random, independent from their attributes, from the positive distribution. The propensity score $e(x)$, which is the probability for selecting a positive example, is constant and equal to the label frequency:

$$e(\boldsymbol{x}) = p(s = 1|\boldsymbol{x}, y = 1) = p(s = 1|y = 1) = c. \tag{4}$$

However, many PU learning applications suffer from labeling bias. The Selected Completely At Random (SCAR) assumption does not conform to realtity. For example, whether someone clicks

on a sponsored search ad is influenced by the position in which it is placed. Similarly, whether a patient with a disease will see a doctor depends on his socio-economic status and the severity of his symptoms. The Selected At Random (SAR) assumption, is the most general assumption about the labeling mechanism: the probability for selecting positive examples to be labeled depends on its attribute values [1].

**Defintion 2.** **SAR**(Selected At Random [3]): The labeling mechanism depends on the values of the attributes of the example. However given the attribute values, it does not depend on the probability of the example being positive. Instead of assuming a constant probability for all positive examples to be labeled, it assumes that the probability is a function of a subset of an example's attributes: $e(\boldsymbol{x}) = p(s = 1|\boldsymbol{x}, y = 1)$.

In biased labeling scenarios, the accuracy of common PU algorithms decreases significantly. So we need to develop the PUe algorithm. The mathematical representation of equation is exactly the same as that of the propensity score in causal inference. Causal inference is the process of determining the actual and independent effects (effects) of a given phenomenon (cause) within a larger system. Causality inference can give evidence of causality model established by causal reasoning. Causal inference is therefore fundamentally a missingdata problem [13], which is very similar to the PU problem. The PU problem is essentially a classification problem of missing negative labels.

What we're actually using here is reverse causal reasoning. In general, we study the average causal effect of a treatment. However, we first complete the causal analysis of the processing (i.e. selection) and the label itself, and then use the known causal analysis, combined with the selected samples and the unselected samples to determine the distribution of positive samples. In this paper, accept processing means that the sample is selected for labeled. And the result is refers to whether the sample itself is a positive sample. Without noise in the detection, whether the sample itself is a positive sample is determined at the time when the sample is generated, and the behavior of selecting is performed after the sample is generated. In causal inference, what happens after the result is produced has no effect on the result, i.e. whether the sample is selected has a causal effect on the label value is 0. In essence, individual causality is also 0. Based on such a simple causality, we use the method of reweighting the sample to estimate the positive probability of the sample.

Since the causal effect of the labeling mechanism on the sample category is zero, known causality can be utilized to improve the performance in the case of biased PU problems. Rubin Causal Model of (RCM) [12] causal inference are extensively used within Statistics, Political Science, Economics, and Epidemiology for reasoning about causation. Due to the presence of confounders, estimating the average treatment effect directly using the average of observed treatment/control outcomes would include a spurious effect, which is called selection bias, brought by the confounders. A key concept in RCM is balancing score [11], reweighting samples by which is one of the most effective methods to overcome selection bias. Besides, propensity score is a special case of balancing score. The propensity score is defined as the conditional probability of treatment given background variables [22]. Propensity score is one of the core concepts of assignment mechanism in RCM. Propensity scores play a central role in observational studies of causality. Propensity scores can be used to reduce selection bias by equating groups based on these covariates, which is called inverse propensity weighting (IPW) [21].

In causal inference, inverse probability weighting (IPW) [21] is a standard statistical technique about propensity score for calculating statistics standardized to a pseudo-population different from that in which the data was collected. Study designs with a disparate sampling population and population of target inference are common in application. Weighting, when correctly applied, can potentially improve the efficiency and reduce the bias of unweighted estimators. In the context of causal inference and survey methodology, propensity scores are estimated (via methods such as logistic regression, random forests, or others), using some set of covariates. These propensity scores are then used as estimators for weights to be used with Inverse probability weighting methods. We incorporate the propensity score when learning in a PU setting by using the propensity scores to reweight the data. The examples are weighted with the inverse of their propensity score.

A crucial difference with the propensity score from causal inference is that our score is conditioned on the class being positive. IPW cannot be applied when working with positive and unlabeled data, because we have zero probability for labeling negative examples. But we can only do a weighting on positive. For each labeled example $(\boldsymbol{x}_i, s = 1)$, which has a propensity score $e_i$, there are expected

to be $\frac{1}{e_i}$ positive examples. So we can use normalized inverse probability weighting on positive to emtimate the unbiased expectations on positive.

$$R_P(g, +1) = \mathbb{E}_{\boldsymbol{x} \sim p_P(\boldsymbol{x})}[L(g(\boldsymbol{x}, +1))] = \mathbb{E}_{c \in (0,1]}\{\mathbb{E}_{\boldsymbol{x} \sim p_P(\boldsymbol{x})}[L(g(\boldsymbol{x}, +1))|e(x) = c]\}. \quad (5)$$

$$\hat{R}_P^e(g, +1) = \sum_{i=1}^{n_P} \frac{1}{\tilde{e}(\boldsymbol{x}_i^P)} L(g(\boldsymbol{x}_i^P), +1). \quad (6)$$

Similarly, $R_P(g, -1) = \mathbb{E}_{\boldsymbol{x} \sim p_P(\boldsymbol{x})}[L(g(\boldsymbol{x}, -1))] = \mathbb{E}_{c \in (0,1]}\{\mathbb{E}_{\boldsymbol{x} \sim p_P(\boldsymbol{x})}[L(g(\boldsymbol{x}, -1))|e(x) = c]\}.$

$$\hat{R}_P^e(g, -1) = \sum_{i=1}^{n_P} \frac{1}{\tilde{e}(\boldsymbol{x}_i^P)} L(g(\boldsymbol{x}_i^P), -1), \quad (7)$$

where $e(\boldsymbol{x}_i^P)$ is the true propensity scores of positive samples and $\hat{e}(\boldsymbol{x}_i^P)$ is the estimate of propensity scores, while $\tilde{e}(\boldsymbol{x}_i^P)$ and $\hat{\tilde{e}}(\boldsymbol{x}_i^P)$ is the normalized inverse probability weighting on positive samples. We can use the real normalized propensity score of positive samples $\tilde{e}(\boldsymbol{x}_i^P)$ to construct the PUe's loss function, or use the estimated normalized propensity score of positive samples $\hat{\tilde{e}}(\boldsymbol{x}_i^P)$ to construct the loss function. Since then, we use $\tilde{e}(\boldsymbol{x}_i^P)$ to represent them.

From the normalization, we can know the following formula: $\sum_{i=1}^{n_P} \frac{1}{\tilde{e}(\boldsymbol{x}_i^P)} = 1, \sum_{i=1}^{n_P} \frac{1}{\hat{\tilde{e}}(\boldsymbol{x}_i^P)} = 1,$

$\frac{1}{\tilde{e}(\boldsymbol{x}_i^P)} = \frac{\frac{1}{e(\boldsymbol{x}_i^P)}}{\sum_{j=1}^{n_P} \frac{1}{e(\boldsymbol{x}_j^P)}}$, and $\frac{1}{\hat{\tilde{e}}(\boldsymbol{x}_i^P)} = \frac{\frac{1}{\hat{e}(\boldsymbol{x}_i^P)}}{\sum_{j=1}^{n_P} \frac{1}{\hat{e}(\boldsymbol{x}_j^P)}}$. Note, $\sum_{j=1}^{n_P} \frac{1}{e(\boldsymbol{x}_j^P)} = N_P$, where $N_P$ means the number of all positive data.

$n_P = n_L$ is the number of labeled samples, $N_P$ is the number of positive samples, $n$ is the number of samples. It is clear that to be unbiased, we have to make $E_{p(x|y=1)} = \frac{n_P}{N_P}$, which indicates that $\sum_{j=1}^{n} e(x) = n_P$. And $E_{p(x|s=1)} \frac{1}{e(x_j^P)} = N_P$. Since $e(x_i^N) > 0$, existing propensity score estimation methods such as SAR-EM in [3] often underestimate the propensity score of positive samples, which will cause co-directional bias or produce an unidentifiable result (overfitting) that all labeled samples have a propensity score close to 1 and all unlabeled samples have a propensity score close to 0.

The normalization technique is used to make the deviations no longer in the same direction (co-directional error will produce worse results). Moreover, the maximum estimation error ratio of sample weights is reduced. The regularization technique is used to alleviate the overfitting problem, and the work does not assume a known propensity score or a positive sample distribution. After normalization is used, $\theta A < \frac{\frac{1}{\hat{e}(x_i^L)}}{\sum_j \hat{e}(x_j^L)} < A/\theta$ does not have such co-direction error, where $\frac{\frac{1}{e(x_i^L)}}{\sum_j e(x_j^L)} = A$ and $\theta < P(y = 1|x_i^L) \leq 1$.

To calculate the unbiased PU loss in the biased scenario, we must know the propensity score of each labeled sample. But this propensity score we usually don't know. So we consider two cases: (1) the true propensity scores we know and (2) we must estimate the propensity scores from data.

**Case 1: Know the True Propensity Scores.** In standard PN Classification, the true PN risk with real class $y$ of size $n$ can be formulated as follows:

$$R_{PN}(g|y) = \pi \hat{R}_P(g, +1) + (1 - \pi)\hat{R}_N(g, -1) = \frac{1}{n} \sum_{i=1}^{n} y_i L(g(\boldsymbol{x}_i), +1) + (1 - y_i)L(g(\boldsymbol{x}_i), -1). \quad (8)$$

**Defintion 3 (IPW).** Given the propensity scores $e$ and PU labels, the inverse probability weighting estimator of $\hat{R}_{PUe}(g)$ is:

$$\hat{R}_{PUe}(g) = \pi \hat{R}_P^e(g, +1) + \hat{R}_U(g, -1) - \pi \hat{R}_P^e(g, -1). \quad (9)$$

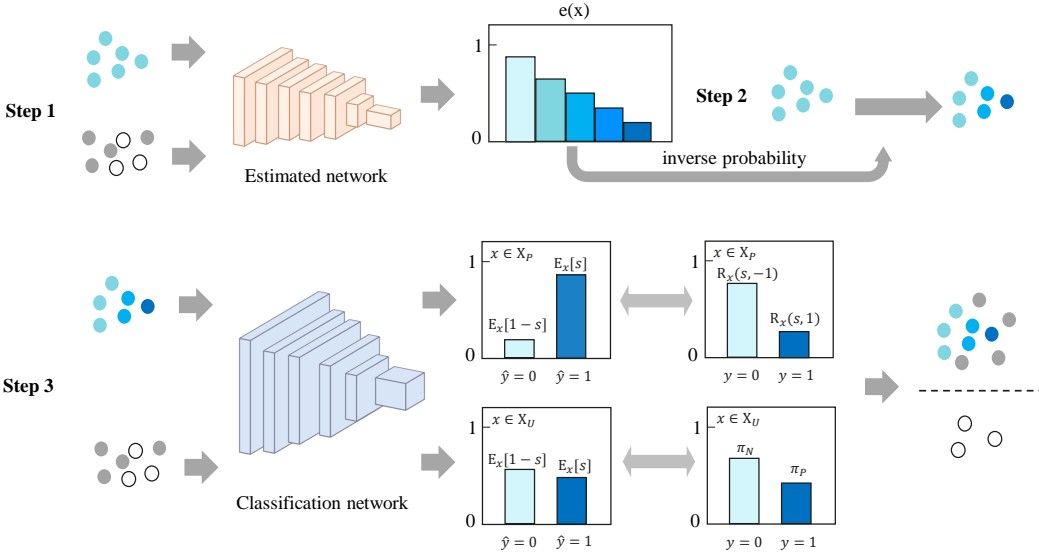

Figure 2: Overview of the PUe framework. Our goal is to estimate the propensity score of the labeled samples and modify the sample weights using the normalized inverse probability weighting technique to obtain the loss function uniformly unbiased estimator.

This estimator is unbiased.

$$\mathbb{E}[\hat{R}_{PUe}(g)] = R_{PN}(g|y) = \frac{1}{n}\sum_{i=1}^{n} y_i L(g(\boldsymbol{x}_i), +1) + (1 - y_i)L(g(\boldsymbol{x}_i), -1). \tag{10}$$

When the propensity score is known, not only the loss function reconstructed by the PUe algorithm is unbiased, the expectation of the PUe loss function is equal to the real PN loss function, but also the difference between the PUe loss function value and the real PN loss function value is probability bounded. However, in other PU algorithms where the label distribution is biased, the expectation of the loss function is biased [3].

**Theorem 1: Inverse Probability Weighting Error Bound.** *For any predicted classes $\hat{y}$ and the real class $y$, with probability $1 - \eta$, the inverse probability weighting estimator $\hat{R}_{PUe}(g)$ does not differ from the true PN loss function $R_{PN}(g|y)$ more than: $|R_{PN}(g|y) - \hat{R}_{PUe}(g)| \leq \sqrt{\frac{L_{\max}^2 \ln \frac{2}{\eta}}{2n}}$. with $L_{\max}$ is the maximum absolute value of cost function $L(g, y)$.*

**Theorem 2: Error Bounds of Common PU Algorithm in Biased Scenarios.** *For any predicted classes $\hat{y}$ and the real class $y$, with probability $1 - \eta$, the inverse probability weighting estimator $\hat{R}_{PU}(g)$ does not differ from the true PN loss function $R_{PN}(g|y)$ more than:*
$|R_{PN}(g|y) - \hat{R}_{PU}(g)| \leq 2\pi L_{\max} * \frac{n_u}{n_u + n_p} + \sqrt{\frac{L_{\max}^2 \ln \frac{2}{\eta}}{2n}}.$
*In the biased scenario, the error bound of PU algorithm is larger than that of PUe algorithm.*

**Case 2: Estimate the Propensity Scores from Data.** But in practice, the probability of the sample being labeled is usually unknown. For example, in ad recommendation, we do not actually know the probability of everyone clicking on the ad, that is, the propensity score is unknown. So in order to get the loss function of PUe algorithm for the unknown label distribution, we must have an estimate $\hat{e}$ of the propensity score.

$$\hat{R}_{PU\hat{e}}(g) = \pi\hat{R}_P^{\hat{e}}(g, +1) + \hat{R}_U(g, -1) - \pi\hat{R}_P^{\hat{e}}(g, -1). \tag{11}$$

In this case, the bias of the inverse probability weighting estimator is:

$$bias(\hat{R}_{PU\hat{e}}(g)) = \pi\sum_{i=1}^{n}y_i\{(\frac{1}{N_P} - \frac{\tilde{e}(\boldsymbol{x}_i)}{N_P\hat{\tilde{e}}(\boldsymbol{x}_i)}) * [L(g(\boldsymbol{x}_i), +1) - L(g(\boldsymbol{x}_i), -1)]\}. \tag{12}$$

From the bias, we can find that the propensity scores only need to be used accurately for positive examples. Apparently, $L(g(\boldsymbol{x}_i), +1) - L(g(\boldsymbol{x}_i), -1) < 0$ for positive examples. And when incorrect propensity score is close to 0 or 1 (especially tend towards 0), there will be a large bias. Overestimate normalized inverse propensity scores.

Underestimated propensity scores are expected to result in a model with a higher bias. Lower propensity scores result in learning models that estimate the positive class to be more prevalent than it is, which results in a larger $[L(g(\boldsymbol{x}_i), +1) - L(g(\boldsymbol{x}_i), -1)]$ for positive samples.

$$\hat{e}(x) = \arg\min_{e} \frac{\pi_1}{n_P}\sum_{i=1}^{n_P}L(e(\boldsymbol{x}_i^P), +1) + \frac{1-\pi_1}{n_U}\sum_{i=1}^{n_U}L(e(\boldsymbol{x}_i^U), -1) + \alpha_e|\sum_{x_i\in\chi_P\cup\chi_U}e(x_i) - n_P|, \tag{13}$$

where $\pi_1 = \frac{n_P}{n_P+n_U}$.

In reality, it is almost impossible to know the propensity score of the sample, which requires us to estimate the propensity score of the sample. However, the traditional statistical estimation method can not be used directly in deep neural network. This requires us to design a new method to estimate sample propensity scores.

According to the paper [7]: In the Local Certainty scenario, we assume the relationship between the observed features and the true class is a deterministic function $f : X \to Y$, where X is the feature space and $Y = \{0, 1\}$, while allowing the propensity score to be an arbitrary probability function. but this assumption is too strong. In fact, we only need to have hypotheses $|P(y = 1|x) - P(y = 0|x)| > \theta$ and $P(y = 1|x^L) > P(y = 0|x^L)$ to estimate the normalized propensity score well for sample classification. (where L represents the labeled sample and is used to distinguish it from the positive category.)

Otherwise, a linear function is usually used to estimate the propensity score in scenarios where the local certainty scenario is not met. Our supplementary experiments in the supplemental materials show that the probability function fitted using deep learning is better than the linear estimation in the MNIST experiment.

We use labeled data as positive samples and unlabeled data as negative samples. We use binary network and regularization technology to construct a scoring network $e(x)$. Finally, the output of the network passes through the sigmoid function to ensure that the output value is between 0 and 1. The score it outputs for each sample is the estimated probability of the sample being labeled (propensity score). The purpose of introducing the regularization technique is to prevent overfitting. Otherwise, the output value of the labeled sample is very close to 1 and the output value of the unlabeled sample is very close to 0. This is not what we expect. The ideal propensity score value should be far away from 0 and 1. Therefore, we introduce such a constraint in the classification loss function by combining the nature of propensity scores, where the sum of propensity scores of all samples is equal to the number of labeled samples: $\alpha_e|\sum_{x_i\in\chi_P\cup\chi_U}e(x_i) - n_P|$.

where $\alpha_e$ ajusts the importance of regular term.

Theoretically, for each labeled example $(\boldsymbol{x}_i, s = +1)$, which has a porpensity score $e_i$, there are expected to be $\frac{1}{e_i}$ positive examples, and of which $\frac{1}{e_i} - 1$ did not get selected to be labeled. Each labeled example gets a weight $\frac{1}{e_i}$ and the negative example is added to the dataset that gets a negative weight $1 - \frac{1}{e_i}$.

According to [7, 3], $P(\hat{Y}|E, L) = \frac{1}{n}\sum_{i=1}^{n}s_i(\frac{1}{e_i}\delta_1 + (1 - \frac{1}{e_i})\delta_0) + (1 - s_i)\delta_0$, using Propensity Sores For Classification. However, as discussed above, existing methods always underestimate

Table 1: Comparative results on MNIST, CIFAR-10 and Alzheimer.

| Dataset | Method | ACC (%) | Prec. (%) | Rec. (%) | F1 (%) | AUC (%) | AP (%) |
|---|---|---|---|---|---|---|---|
| MNIST | uPU | 85.17 (1.90) | 78.38 (2.36) | 96.69 (1.33) | 86.55 (1.50) | 85.34 (1.87) | 77.41 (2.26) |
| | nnPU | 87.81 (1.65) | 83.78 (2.14) | 93.40 (1.68) | 88.31 (1.51) | 87.89 (1.63) | 81.62 (2.28) |
| | PUbN | 87.07 (1.73) | 83.43 (2.74) | 92.21 (2.55) | 87.55 (1.55) | 87.14 (2.74) | 80.70 (2.59) |
| | Dist-PU | 89.79 (0.59) | 92.22 (1.67) | 88.86 (1.48) | 90.49 (0.47) | 96.60 (0.30) | 96.92 (0.21) |
| | uPUe | 92.28 (1.32) | 89.41 (1.87) | 95.71 (1.43) | 92.44 (1.25) | 92.33 (1.31) | 87.69 (1.92) |
| | nnPUe | 92.45 (1.61) | 90.45 (2.26) | 94.73 (1.24) | 92.53 (1.55) | 92.48 (1.60) | 88.29 (2.43) |
| | PUbNe | 94.70 (0.63) | 97.16 (1.59) | 91.33 (1.58) | 94.46 (0.71) | 94.66 (0.50) | 93.18 (0.55) |
| | Dist-PUe | 92.57 (0.78) | 93.02 (0.86) | 90.58 (1.47) | 92.77 (0.84) | 97.13 (0.57) | 97.43 (0.44) |
| | uPUe' | 92.39 (1.36) | 89.33 (1.87) | 96.07 (1.70) | 92.56 (1.30) | 92.44 (1.87) | 87.76 (1.93) |
| | nnPUe' | 92.94 (1.27) | 91.11 (1.84) | 94.98 (2.38) | 92.98 (1.29) | 92.96 (1.27) | 89.00 (1.77) |
| | PUbNe' | 94.48 (0.93) | 97.74 (0.46) | 90.90 (2.10) | 94.18 (1.06) | 94.43 (0.95) | 93.45 (1.00) |
| | Dist-PUe' | 91.73 (0.95) | 93.34 (0.42) | 89.60 (2.38) | 91.41 (1.11) | 96.48 (1.54) | 96.60 (1.37) |
| CIFAR-10 | uPU | 76.33 (1.76) | 83.59 (0.65) | 50.81 (5.57) | 63.01 (4.35) | 84.40 (3.12) | 78.68 (2.22) |
| | nnPU | 82.47 (0.69) | 75.30 (1.35) | 83.68 (1.64) | 79.25 (0.75) | 90.01 (0.73) | 85.22 (0.73) |
| | PUbN | 84.54 (0.54) | 82.18 (1.38) | 78.40 (2.70) | 80.20 (0.99) | 91.62 (0.27) | 87.31 (0.40) |
| | Dist-PU | 84.45 (1.00) | 80.90 (1.51) | 80.04 (1.62) | 80.46 (1.26) | 90.89 (1.20) | 85.67 (1.54) |
| | uPUe | 79.31 (0.90) | 81.37 (2.16) | 62.86 (5.02) | 70.73 (2.58) | 86.76 (0.82) | 80.07 (1.56) |
| | nnPUe | 83.33 (1.04) | 76.83 (2.75) | 83.88 (2.86) | 80.11 (0.85) | 90.53 (0.79) | 85.33 (1.36) |
| | PUbNe | 85.66 (0.58) | 83.18 (1.63) | 80.52 (2.29) | 81.79 (0.66) | 92.69 (0.74) | 89.41 (1.24) |
| | Dist-PUe | 86.97 (0.62) | 82.11 (1.69) | 86.46 (2.00) | 84.19 (0.76) | 93.50 (0.66) | 89.38 (1.78) |
| | uPUe' | 79.11 (1.75) | 85.11 (1.58) | 57.92 (4.96) | 68.80 (3.61) | 86.57 (1.56) | 81.56 (2.18) |
| | nnPUe' | 84.24 (0.67) | 78.40 (0.86) | 83.66 (2.35) | 80.92 (1.02) | 91.35 (0.99) | 87.15 (1.83) |
| | PUbNe' | 87.02 (0.47) | 83.84 (1.44) | 83.78 (2.68) | 83.76 (0.82) | 93.80 (0.42) | 90.69 (0.65) |
| | Dist-PUe' | 86.17 (2.52) | 80.97 (3.20) | 85.57 (3.32) | 83.19 (3.06) | 92.65 (2.20) | 88.22 (3.19) |
| Alzheimer | uPU | 63.46 (1.72) | 66.80 (3.45) | 55.06 (9.83) | 59.65 (4.70) | 68.08 (2.16) | 67.78 (2.22) |
| | nnPU | 69.19 (0.85) | 65.46 (1.06) | 81.46 (5.22) | 72.48 (1.62) | 73.50 (0.84) | 68.74 (1.83) |
| | PUbN | 68.75 (0.96) | 66.65 (2.04) | 75.64 (5.58) | 70.68 (1.52) | 72.57 (0.63) | 66.99 (0.80) |
| | Dist-PU | 69.74 (0.50) | 67.74 (1.57) | 75.69 (4.72) | 71.37 (1.33) | 74.34 (1.10) | 69.53 (1.39) |
| | uPUe | 65.30 (3.36) | 68.53 (1.99) | 56.73 (11.93) | 61.32 (7.58) | 70.49 (3.99) | 70.28 (3.09) |
| | nnPUe | 69.38 (0.56) | 65.97 (0.68) | 80.06 (2.77) | 72.30 (0.91) | 74.49 (0.34) | 69.98 (0.60) |
| | PUbNe | 69.52 (0.85) | 68.28 (1.28) | 72.95 (1.99) | 70.51 (0.82) | 74.68 (1.08) | 70.33 (0.98) |
| | Dist-PUe | 70.51 (0.62) | 68.91 (1.00) | 75.02 (2.26) | 71.81 (0.83) | 75.19 (0.69) | 71.23 (2.08) |

propensity scores. As a result, PU learning classification is performed by using the foregoing formula, which reduces model classification performance.

We found that in cost-sensitive PU learning technique, the training instances are properly reweighted. and a common way to score propensity is inverse probability weighting (a reweighting technique). Conditionally, the two methods can be combined to solve the defects of the two methods themselves. The disadvantage of the cost-sensitive PU learning technique is that the classification effect of the application deteriorates when the labeled samples are biased. The propensity score is estimated by the fact that the propensity score of positive samples is always underestimated, which leads to the decline of the classification effect of the model. This is why we chose to couple with cost-sensitive PU learning.

However, in practice, the disturbance of propensity score has a great influence on the results due to the reciprocal, which makes the variance of our model large. We suggest that the existing prior class $\pi$ in PU algorithm is used, and the propensity score of samples is weighted by normalized inverse probability, which is used as the weight of samples. This advantage is that there are many estimation methods for the prior class [2], and that the error estimation of the propensity score of individual samples is prevented from greatly affecting the model (the propensity score is close to 0), thereby enhancing the robustness of the model and improving the accuracy.The loss function of the basic PUe algorithm can be obtained as follows:

$$\hat{R}_{PU\hat{e}}(g) = \pi\hat{R}_P^{\hat{e}}(g, +1) + \hat{R}_U(g, -1) - \pi\hat{R}_P^{\hat{e}}(g, -1). \tag{14}$$

The main algorithm process of PUe is shown in Figure2. The specific algorithm process of PUe is shown in supplementary material. In summary, the first step of PUe algorithm is to estimate the propensity score of positive samples using regular binary classification. If the propensity score is

known, the first step can be skipped. The second step is to modify positive sample weights with normalized propensity scores. The third step is to use the PU algorithm (that is, PUe) with modified weights to perform PU binary classification. We also consider the case where negative classes are also selectively labeled and conduct related experiments (see PUbN), which are not mentioned in other papers estimating propensity scores and applying them to PU learning.

## 3 Experiment

In this section, we experimented our idea on several popular PU methods and compare its performance against the initial baseline methods.

Table 2: Ablation results on CIFAR-10 with ✓ indicating the enabling of the corresponding loss term and different labeled distripution.

| method | $\alpha_e$ | labeled distripution | ACC (%) | Prec. (%) | Rec. (%) | F1 (%) | AUC (%) | AP (%) |
|---|---|---|---|---|---|---|---|---|
| Dist-PUe | ✓ | [.25,.25,.25,.25] | 91.79 (0.86) | 89.02 (2.63) | 90.78 (1.81) | 89.84 (0.92) | 97.11 (0.52) | 95.38 (1.18) |
| Dist-PUe | | [.25,.25,.25,.25] | 91.30 (0.80) | 89.22 (1.56) | 89.05 (1.93) | 89.11 (1.03) | 96.64 (0.53) | 94.85 (0.77) |
| Dist-PU | | [.25,.25,.25,.25] | 91.88 (0.52) | 89.87 (1.09) | 89.84 (0.81) | 89.85 (0.62) | 96.92 (0.45) | 95.49 (0.72) |
| Dist-PUe | ✓ | [.10,.10,.30,.50] | 91.50 (0.32) | 89.07 (1.15) | 89.78 (0.97) | 89.41 (0.34) | 96.67 (0.14) | 95.64 (0.26) |
| Dist-PUe | | [.10,.10,.30,.50] | 90.40 (0.66) | 88.86 (3.15) | 87.14 (0.74) | 87.89 (0.74) | 95.71 (0.40) | 94.19 (0.65) |
| Dist-PU | | [.10,.10,.30,.50] | 90.79 (0.72) | 91.31 (1.54) | 85.12 (2.44) | 88.07 (1.07) | 95.81 (0.83) | 91.28 (2.35) |
| Dist-PUe | ✓ | [.72,.15,.10,.03] | 86.97 (0.62) | 82.11 (1.69) | 86.46 (2.00) | 84.19 (0.76) | 93.50 (0.66) | 89.38 (1.78) |
| Dist-PUe | | [.72,.15,.10,.03] | 86.44 (0.67) | 82.19 (1.54) | 84.45 (1.78) | 83.28 (0.82) | 92.90 (0.63) | 88.39 (1.32) |
| Dist-PU | | [.72,.15,.10,.03] | 84.45 (1.00) | 80.90 (1.51) | 80.04 (1.62) | 80.46 (1.26) | 90.89 (1.20) | 85.67 (1.54) |
| Dist-PUe | ✓ | [.08,.86,.02,.04] | 88.15 (1.22) | 88.05 (2.69) | 81.70 (5.41) | 84.56 (2.23) | 94.46 (1.01) | 92.79 (1.09) |
| Dist-PUe | | [.08,.86,.02,.04] | 86.39 (2.11) | 84.47 (3.51) | 81.05 (4.32) | 82.62 (2.77) | 92.42 (2.25) | 89.84 (2.40) |
| Dist-PU | | [.08,.86,.02,.04] | 86.74 (1.58) | 86.56 (2.51) | 79.30 (4.60) | 82.65 (2.48) | 93.30 (1.32) | 91.08 (1.69) |

### 3.1 Experimental settings

**Datasets.** We conducted experiments on two benchmarks commonly used in PU learning: MNIST for parity classification and CIFAR-10 [17] for vehicle class recognition. And on the simulated datasets of MNIST and CIFAR-10, we know the propensity score of the sample a priori, and we compare our proposed method with the ideal propensity to know the propensity score.Moreover, we tested our method on the Alzheimer's dataset [2] used to identify Alzheimer's disease in order to test the performance of our proposed method in real-world scenarios. More information is displayed in supplementary material.

**Baselines.** We mainly consider four common PU learning methods, including uPU, nnPU, PUbN, and Dist-PU.

**Evaluation metrics.** For each model, we counted six commonly used indicators of the test set results, including accuracy (ACC), Precision (Prec.), Recall (Rec.), F1, Area Under ROC Curve (AUC) and Average Precision (AP). Accuracy (ACC) is the main index for a more comprehensive comparison. The experiments are repeated with six random seeds, and the mean and standard deviation of each index were recorded.

**Implementation details.** All the experiments are run by PyTorch. Backbones of each dataset are summarized in supplementary material. The training batch size is set as 256 for MNIST and CIFAR-10, while 128 for Alzheimer. We use Adam as the optimizer with a cosine annealing scheduler, where the initial learning rate is set as $5 \times 10^{-3}$; while weight decay is set as $5 \times 10^{-3}$. PU learning methods first experiences a warm-up phase of 60 epochs, and then trains another 60 epochs with depth, where the value of $\alpha$ is searched in the range of $[0, 20]$.

---

[2]Dubey, S. Alzheimer's Dataset. Available online: https://www.kaggle.com/tourist55/ alzheimers-dataset-4-class-of-images

### 3.2 Comparision with state-of-the-art methods

**Competitors.** We compare our method with 4 competitive baseline PU algorithms including uPU [5], nnPU [16], PUbN [9], and Dist-PU [24]. Due to the space limitation, the detailed descriptions are provided in supplementary material.

**Results.** The results on all the datasets are recorded in Tab.1. It shows that in most metrics, our proposed PU+e method significantly outperforms our competitors on all biased datasets, improving the performance of the original PU method by about 1% to 5%. This proves the effectiveness of our proposed method. In addition, some observations can be made: (1) A model with a known propensity score is not necessarily the best (who has e' in Tab1), but a model with an estimated propensity score can perform better in many cases. This is a conclusion that is consistent with causal inference[8]. (2) In the ablation study, the performance of the PU+e algorithm is comparable to that of the most advanced PU algorithm, even when the labels are evenly distributed.

### 3.3 Ablation studies

**Effectiveness of hyper-parameters.** Ablation experiments were conducted to verify the validity of hyperparameters, which is shown in Figure3. It can be seen that the PUe algorithm is sensitive to hyperparameters and does not change monotonically. When $\alpha_e = 15$, the PUe algorithm has the best performance. Therefore, in order to obtain the best performance of the PUe algorithm, fine adjustment is required. When the best performance is achieved remains to be further studied. However, the overall performance of Cifar-10 is better than that of the original Dist-PU algorithm.

**Effectiveness of labeled samples distribution.** The deviation of label data distribution will significantly affect the promotion effect of our proposed model in Tab.2. In general, when the distribution of labeled samples is more biased, the improvement effect of our proposed method is more obvious. However, when the distribution deviation is very large, the lifting effect of our model is weakened. This is because the number of labeled samples for some classes is too small or not labeled. As a result, the propensity score is close to 0, which affects the model performance. On the Cifar10 dataset, our algorithm improves the accuracy of the Dist-PU algorithm by up to 2.5%.

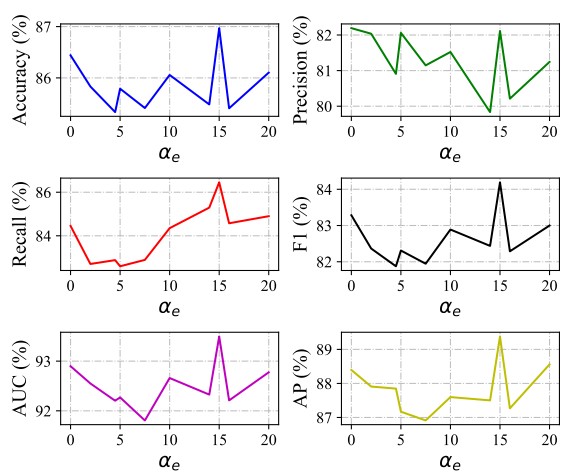

Figure 3: Influences of different $\alpha_e$ on CIFAR-10 by Dist-PUe.

## 4 Conclusion

In this paper, a new PU learning method, namely PUe, is proposed from the perspective of propensity score for the PU learning with biased labels in deep learning. The PUe algorithm can enhance the original cost-sensitive PU algorithm, improve the prediction precision in the case of biased sample labeling, and has the degenerate ability, and the prediction precision in the case of unbiased labeling is not lower than that of the original PU algorithm. PUe consistently outperforms state-of-the-art methods on most metrics on real-world biased labeled datasets, including MNIST, CIFAR-10, and Alzheimer's. We hope that the proposed propensity score estimation scheme for deep learning can also provide some inspiration for other weakly supervised scenarios, especially those where the label distribution is unknown.

## Acknowledgement

We gratefully acknowledge the support of MindSpore [20], CANN (Compute Architecture for Neural Networks) and Ascend AI Processor used for this research.

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

# A  Algorithm

---

**Algorithm 1** PUe algorithm

---

**Require:** data $\chi_P, \chi_U$, size $n, n_P, n_U$, hyperparameter $\alpha_e, \pi$.
  1: **Step 1:**
  2: Compute $\hat{e}(x)$ by minimizing $\frac{\pi}{n_P} \sum_{i=1}^{n_P} L(g(\boldsymbol{x}_i^P), +1) + \frac{1-\pi}{n_U} \sum_{i=1}^{n_U} L(g(\boldsymbol{x}_i^U), -1) + \alpha_e |\sum_{x_i \in \chi_P \cup \chi_U} e(x_i) - n_P|$;
  3: **Step 2:**
  4: Compute the weight of labeled samples: $w_i^P = \frac{\pi}{\hat{e}(x_i^P)}$
  5: **Step 3:**
  6: **for** $i = 1 \ldots$ **do**
  7:     Shuffle $(\chi_P, \chi_U)$ into M mini-batches
  8:     **for** each mini-batch $(\chi_P^j, \chi_U^j)$ **do**
  9:         Compute the corresponding $\hat{R}_{\text{PUe}}(g)$
 10:         Use $\mathcal{A}$ to update $\theta$ with the gradient information $\nabla_\theta \hat{R}_{\text{PUe}}(g)$
 11:     **end for**
 12: **end for**

---

# B  Experiment Details

Table 3: Summary of used datasets and their corresponding models.

| Dataset | Input Size | $n_P$ | $n_U$ | # Testing | $\pi_P$ | Positive Class | true e(x) | Model |
|---|---|---|---|---|---|---|---|---|
| MNIST | $28 \times 28$ | 2500 | 60,000 | 10,000 | 0.5 | Even (0, 2, 4, 6 and 8) | [.65,.15,.1,.07,.03] | 6-layer MLP |
| CIFAR-10 | $3 \times 32 \times 32$ | 1,000 | 50,000 | 10,000 | 0.4 | Vehicles (0, 1, 8 and 9) | [.72,.15,.1,.03] | 13-layer CNN |
| Alzheimer | $3 \times 224 \times 224$ | 769 | 5,121 | 1,279 | 0.5 | Alzheimer's Disease | unknow | ResNet-50 |

# C  Complementary Experiment

Table 4: Supplemental Experiments on MINST

| method | labeled distripution | ACC.(%) | Prec.(%) | Rec.(%) | F1.(%) | AUC.(%) | AP.(%) |
|---|---|---|---|---|---|---|---|
| LRe | [.65,.15,.10,.07,.03] | 86.19(0.75) | 92.94(0.64) | 77.89(1.38) | 84.75(0.93) | 88.06(0.93) | 88.72(1.09) |
| nnPUe | [.65,.15,.10,.07,.03] | 92.45 (1.61) | 90.45 (2.26) | 94.73 (1.24) | 92.53 (1.55) | 92.48 (1.60) | 88.29 (2.43) |
| nnPU without normalize | [.65,.15,.10,.07,.03] | 90.95(1.61) | 88.18(2.40) | 94.38(2.74) | 91.13(1.56) | 91.00(1.61) | 85.98(2.25) |
| Self-PU | [.65,.15,.10,.07,.03] | 90.08(0.47) | 90.08 (0.47) | 89.35 (1.17) | 90.70 (1.73) | 90.00 (0.53) | 85.61 (0.69) |
| anchor | [.65,.15,.10,.07,.03] | 88.22(0.95) | 94.66(1.41) | 80.70(3.15) | 87.06(1.37) | 92.36(1.92) | 93.37(2.33) |

LRe: Logistic regression estimation of propensity scores for PU learning.

According to paper [7], it cannot estimate identifiable PS without making certain assumptions about the data. But according to the formula we gave in the first question, it's approximate. This is not explained by the self-monitoring method. Results in the above table show that our scheme is better than self-PU in the case of biased label datasets.

# D  Proofs

## D.1  error bound of bias

We may assume that the error of propensity scores estimated by the NN method is the same as that estimated by the linear method. (In fact, the NN methods are usually more general, which may produce results with less error.) That is, the estimate of the propensity score has a maximum error ratio of $\beta$, with $\beta e(x_i^L) \leq \hat{e}(x_i^L) \leq e(x_i^L)$. of the following shows that our regularization technique can yield a smaller error ratio with respect to sample weights. Obviously, the sample $x_i^L$ has a sample weight of $\frac{1}{n\hat{e}(x_i^L)}$. in (Formula 1) with an error bound of $bias(\frac{1}{n\hat{e}(x_i^L)}) \leq \frac{1}{ne(x_i^L)}(\frac{1}{\beta} - 1)$.

## D.2 error ratio

In our approach, Sample $x_i^L$ has a weight of $\pi \frac{\frac{1}{\hat{e}(x_i^L)}}{\sum_j \frac{1}{\hat{e}(x_j^L)}} . P(\gamma e(x_i^L) < \hat{e}(x_i^L) \leq e(x_i^L)) = \alpha$ where the set of samples is $S_1$ . $P(\beta e(x_i^L) < \hat{e}(x_i^L) \leq \gamma e(x_i^L)) = 1 - \alpha$ where the set of samples is $S_2$. $\beta < \gamma < 1$ and $\sum_{i \in S_1} \frac{1}{e(x_i^L)} = \sum_{i \in S_2} \frac{1}{e(x_i^L)} = B$. So that $\sum_j \frac{1}{e(x_j^L)} = 2B = N_p$. For $x_i^L \in S_1$, we have $\frac{1}{e(x_i^L)} \leq \frac{1}{\hat{e}(x_i^L)} < \frac{1}{\gamma e(x_i^L)}$. For $x_i^L \in S_2$, we have $\frac{1}{\gamma e(x_i^L)} \leq \frac{1}{\hat{e}(x_i^L)} < \frac{1}{\beta e(x_i^L)}$, so we can get $B(1 + \frac{1}{\gamma}) \leq \sum_j \frac{1}{\hat{e}(x_j^L)} < B(\frac{1}{\gamma} + \frac{1}{\beta})$ and we have $bias(\pi \frac{\frac{1}{\hat{e}(x_i^L)}}{\sum_j \frac{1}{\hat{e}(x_j^L)}}) \leq \max[\frac{1}{ne(x_i^L)}(\frac{2}{(1+\gamma)} - 1), \frac{1}{ne(x_i^L)}(1 - \frac{2}{\frac{1}{\beta} + \frac{1}{\gamma}}), \frac{1}{ne(x_i^L)}(\frac{2}{(1+\frac{1}{\gamma})\beta} - 1), \frac{1}{ne(x_i^L)}(1 - \frac{2}{\frac{2}{\beta} + 1})] < \frac{1}{ne(x_i^L)}(\frac{1}{\beta} - 1)$ and obviously we have $\frac{2}{(1+\gamma)} < \frac{2}{(1+\frac{1}{\gamma})\beta} < \frac{1}{\beta}, 0 < 1 - \frac{2}{\frac{\gamma}{\beta}+1} < 1 - \frac{2}{\frac{1}{\beta}+\frac{1}{\gamma}} < 1 - \beta < \frac{1}{\beta} - 1$, which shows that our regularization technique has a smaller error ratio with respect to sample weights.

## D.3 expectation

One understanding is that, according to the PS definition, each labeled sample $x_j^P$ corresponds to $\frac{1}{e(x_j^P)}$ positive samples. So $\sum_{j=1}^{n_p} \frac{1}{e(x_j^P)} = N_p$ it's true. Because $P(x|s=1) = P(x, y = 1|s = 1)$, we have

$E_{P(x|s=1)} \frac{1}{P(s=1|x,y=1)}$

$= \sum P(x, y = 1|s = 1) \frac{1}{P(s=1|x,y=1)}$

$= \sum \frac{P(s=1|x,y=1)P(x,y=1)}{P(s=1)} \frac{1}{P(s=1|x,y=1)}$

$= \sum \frac{P(x,y=1)}{P(s=1)} = \frac{n}{n_P} \sum P(x, y = 1)$

$= \frac{n}{n_P} \frac{N_P}{n} = \frac{N_P}{n_P}$.

It indicates that $\sum_{j=1}^{n_p} \frac{1}{e(x_j^P)} = N_p$.

## D.4 PUbN

The PUbN formula is as follows:

Let $\sigma(x) = p(s = +1|x)$, however, the $\sigma(x)$ is actually unknown, we should replace $\sigma(x)$ by its estimate $\hat{\sigma}(x)$. We can get the classification risk of PUbN $(R_{PUbN}(g))$, as the following expression:

$R_{PUbN}(g) = \pi R_P(g, +1) + \rho R_{bN}(g, -1) + \bar{R}_{s=-1,\eta,\hat{\sigma}}(g)$

where $\bar{R}_{s=-1,\eta,\hat{\sigma}}(g) = \mathbb{E}_{x \sim p(x)}[\mathbb{1}_{\hat{\sigma}(x) \leq \eta} L(-g(x))(1 - \hat{\sigma}(x))]$ $+ \pi \mathbb{E}_{x \sim p_P(x)}[\mathbb{1}_{\hat{\sigma}(x) > \eta} L(-g(x)) \frac{1-\hat{\sigma}(x)}{\hat{\sigma}(x)}] + \rho \mathbb{E}_{x \sim p_{bN}(x)}[\mathbb{1}_{\hat{\sigma}(x) > \eta} L(-g(x)) \frac{1-\hat{\sigma}(x)}{\hat{\sigma}(x)}]$

Then $R_{bN}(g, -1)$ and $\bar{R}_{s=-1,\eta,\hat{\sigma}}(g)$ can also be approximated from data by $\hat{R}_{bN}(g, -1) = \frac{1}{n_{bN}} \sum_{i=1}^{n_{bN}} L(g(x_i^{bN}), -1)$ $\hat{\bar{R}}_{s=-1,\eta,\hat{\sigma}}(g) = \frac{1}{n_U} \sum_{i=1}^{n_U} [\mathbb{1}_{\hat{\sigma}(x_i^U) \leq \eta} L(g(x_i^U), -1)(1 - \hat{\sigma}(x_i^U))]$ $+ \frac{\pi}{n_P} \sum_{i=1}^{n_P} [\mathbb{1}_{\hat{\sigma}(x_i^P) > \eta} L(g(x_i^P), -1) \frac{1-\hat{\sigma}(x_i^P)}{\hat{\sigma}(x_i^P)}] + \frac{\rho}{n_{bN}} \sum_{i=1}^{n_{bN}} [\mathbb{1}_{\hat{\sigma}(x_i^{bN}) > \eta} L(g(x_i^{bN}), -1) \frac{1-\hat{\sigma}(x_i^{bN})}{\hat{\sigma}(x_i^{bN})}]$

$\hat{R}_{PUbN,\eta,\hat{\sigma}}(g) = \pi \hat{R}_P(g, +1) + \rho \hat{R}_{bN}(g, -1) + \hat{\bar{R}}_{s=-1,\eta,\hat{\sigma}}(g)$

## D.5 PUbNe

Our PUbNe formula is as follows:

$\hat{R}_{PUbN\hat{e},\eta,\hat{\sigma}}(g) = \pi \hat{R}_P^{\hat{e}}(g, +1) + \rho \hat{R}_{bN}^{\hat{e}}(g, -1) + \hat{\bar{R}}_{s=-1,\eta,\hat{\sigma}}^{\hat{e}}(g)$

,where $\hat{R}_{bN}^{\hat{e}}(g, -1) = \sum_{i=1}^{n_{bN}} \frac{1}{\hat{e}(x_i^{bN})} L(g(x_i^{bN}), -1)$ and

$$\hat{\hat{R}}^{\hat{e}}_{s=-1,\eta,\hat{\sigma}}(g) = \frac{1}{n_U}\sum_{i=1}^{n_U}[\mathbb{1}_{\hat{\sigma}(x_i^U)\leq\eta}L(g(x_i^U),-1)(1 - \hat{\sigma}(x_i^U))]$$
$$+\pi\sum_{i=1}^{n_P}[\frac{1}{\hat{e}(x_i^P)}\mathbb{1}_{\hat{\sigma}(x_i^P)>\eta}L(g(x_i^P),-1)\frac{1-\hat{\sigma}(x_i^P)}{\hat{\sigma}(x_i^P)}]+\rho\sum_{i=1}^{n_{bN}}[\frac{1}{\hat{e}(x_i^{bN})}\mathbb{1}_{\hat{\sigma}(x_i^{bN})>\eta}L(g(x_i^{bN}),-1)\frac{1-\hat{\sigma}(x_i^{bN})}{\hat{\sigma}(x_i^{bN})}]$$

### D.6 unbiased

$\mathbb{E}[\hat{R}_{PUe}(g)]$

$$= \mathbb{E}[\pi\hat{R}_P^e(g,+1) + \hat{R}_U(g,-1) - \pi\hat{R}_P^e(g,-1)]$$

$$= \mathbb{E}[\frac{1}{n}\sum_{i=1}^{n_P}\frac{1}{e(\boldsymbol{x}_i^P)}\left(L(g(\boldsymbol{x}_i^P),+1) - L(g(\boldsymbol{x}_i^P),-1)\right) + \frac{1}{n}\sum_{i=1}^{n}L(g(\boldsymbol{x}_i),-1)]$$

$$= \mathbb{E}[\frac{1}{n}\sum_{i=1}^{n_P}\frac{1}{e(\boldsymbol{x}_i^P)}L(g(\boldsymbol{x}_i^P),+1) + \left(1 - \frac{1}{e(\boldsymbol{x}_i^P)}\right)L(g(\boldsymbol{x}_i^P),-1) + \frac{1}{n}\sum_{i=1}^{n}(1-s_i)L(g(\boldsymbol{x}_i),-1)]$$

$$= \mathbb{E}[\frac{1}{n}\sum_{i=1}^{n}s_i\frac{1}{e(\boldsymbol{x}_i^P)}L(g(\boldsymbol{x}_i^P),+1) + s_i\left(1 - \frac{1}{e(\boldsymbol{x}_i^P)}\right)L(g(\boldsymbol{x}_i^P),-1) + (1-s_i)L(g(\boldsymbol{x}_i),-1)]$$

$$= \frac{1}{n}\sum_{i=1}^{n}y_ie_i\frac{1}{e(\boldsymbol{x}_i)}L(g(\boldsymbol{x}_i),+1) + y_ie_i\left(1 - \frac{1}{e(\boldsymbol{x}_i)}\right)L(g(\boldsymbol{x}_i),-1) + (1-y_ie_i)L(g(\boldsymbol{x}_i),-1)$$

$$= \frac{1}{n}\sum_{i=1}^{n}y_iL(g(\boldsymbol{x}_i),+1) + y_i\left(e_i - 1\right)L(g(\boldsymbol{x}_i),-1) + (1-y_ie_i)L(g(\boldsymbol{x}_i),-1)$$

$$= \frac{1}{n}\sum_{i=1}^{n}y_iL(g(\boldsymbol{x}_i),+1) + (1-y_i)L(g(\boldsymbol{x}_i),-1)$$

$$= R_{PN}(g|y).$$

$$(15)$$

The change of $\hat{R}_{PUe}(g)$ will be no more than $L_{\max}/n$ if some $x_i \in \chi_P \cup \chi_U$ is replaced, and McDiarmid's inequality gives us:

$$\Pr\{|\hat{R}_{PUe}(g) - R_{PN}(g|y)| \geq \epsilon\} = \Pr\{|\hat{R}_{PUe}(g) - \mathbb{E}[\hat{R}_{PUe}(g)]| \geq \epsilon\} \leq 2\exp\left(-\frac{2\epsilon^2}{n(L_{\max}/n)^2}\right).$$

And make the right side of the previous formula equal to $\eta$:

$$2\exp\left(-\frac{2\epsilon^2}{n(L_{\max}/n)^2}\right) = \eta$$

$$\Longleftrightarrow \exp\left(\frac{2\epsilon^2}{n(L_{\max}/n)^2}\right) = \frac{2}{\eta}$$

$$\Longleftrightarrow \frac{2\epsilon^2}{L_{\max}^2/n} = \ln\left(\frac{2}{\eta}\right)$$

$$\Longleftrightarrow 2\epsilon^2 = \frac{L_{\max}^2\ln\left(\frac{2}{\eta}\right)}{n}$$

$$\Longleftrightarrow \epsilon = \sqrt{\frac{L_{\max}^2\ln\left(\frac{2}{\eta}\right)}{2n}}$$

Equivalently, with probability at least $1 - \eta$,

$$|\hat{R}_{PUe}(g) - R_{PN}(g|y)| = |\hat{R}_{PUe}(g) - \mathbb{E}[\hat{R}_{PUe}(g)]| \leq \sqrt{\frac{L_{\max}^2\ln\frac{2}{\eta}}{2n}}. \qquad (16)$$

And because we know the expressions of $\hat{R}_{PUe}(g)$ and $\hat{R}_{uPU}(g)$:

$$\hat{R}_{PUe}(g) = \pi\hat{R}_P^e(g,+1) + \hat{R}_U(g,-1) - \pi\hat{R}_P^e(g,-1), \tag{17}$$

$$\hat{R}_{PU}(g) = \pi\hat{R}_P(g,+1) + \hat{R}_U(g,-1) - \pi\hat{R}_P(g,-1), \tag{18}$$

Since we know that $\sum_{j=1}^{n_P} \frac{1}{e(\boldsymbol{x}_j^P)} = N_P$, we can get the following formula:

$$|\hat{R}_{PUe}(g) - \hat{R}_{PU}(g)|$$

$$= \pi|\hat{R}_P^e(g,+1) - \hat{R}_P^e(g,-1) - \left((\hat{R}_P(g,+1) - \hat{R}_P(g,-1)\right)|$$

$$\leq \frac{1}{n}|\sum_{i=1}^{n_P}(\frac{1}{e(\boldsymbol{x}_i^P)} - \frac{N_P}{n_p})L(g(\boldsymbol{x}_i^P),+1)| + \frac{1}{n}|\sum_{i=1}^{n_P}(\frac{1}{e(\boldsymbol{x}_i^P)} - \frac{N_P}{n_p})L(g(\boldsymbol{x}_i^P),-1)|$$

$$\leq \frac{2}{n}N_P L_{\max} = 2\pi L_{\max} \tag{19}$$

Then, we can prove:

$$|R_{PN}(g|y) - \hat{R}_{PU}(g)|$$

$$\leq |R_{PN}(g|y) - \hat{R}_{PUe}(g)| + |\hat{R}_{PUe}(g) - \hat{R}_{PU}(g)|$$

$$\leq 2\pi L_{\max} + \sqrt{\frac{L_{\max}^2 \ln \frac{2}{\eta}}{2n}}. \tag{20}$$

