# OpenReview forum: "PUe: Biased Positive-Unlabeled Learning Enhancement by Causal Inference"
_NeurIPS.cc/2023/Conference — NeurIPS 2023 poster_

### Official Review · Reviewer_w1uD · 2023-06-10

**Soundness:** 1 poor
**Presentation:** 1 poor
**Contribution:** 2 fair
**Rating:** 4
**Confidence:** 4

**Summary:**

This paper works on PU learning with a selection bias. They propose a weighted risk estimator to solve this problem, and estimate the weight via neural networks. Extensive experimental results validate the effectiveness of the proposed approach.

**Strengths:**

- The studied problem is very important for PU learning.
- The experimental results is good, since the proposed mechanism improves previous PU learning methods under the biased setting.

**Weaknesses:**

- First, I found potential mistakes in the derivations of  $\mathbb{E}[\hat{R}\_{PUe}(g)]=R_{PN}(g|y)$ in the supplemental materials. In Line 10 of Section 3, the authors use $n_{P}/n$ to replace $\pi$, but I do not think so. Since there are positive examples in the unlabeled data ($s=0, y=1$), the class prior is under estimated. I will reconsider it if I need to be corrected.

- Second, the major problem of this paper is that it is very similar to [3]. The proposed approach and theoretical results are very similar. For the theoretical results, Theorem 1 and 2 are very similar to Proposition 3 and 4 in [3]. For the methodology, it only changes the MLE method in [3] to a neural network. Therefore, the contribution is limited.

- Third, this work actually works on “single-training-set PU learning”, while uPU and nnPU work on "case-control PU learning". Please see Gong et al. (2021) for more explanations. The authors should discuss the problem here.

- Besides, I also have questions on the weight estimation via a neural network. The paper says that it regards positive examples as "positive" and unlabeled examples as "negative". My concern is that the model output is $p(s=1|x)$, instead of $p(s=1|y=1,x)$. Can the authors explain it?

- It seems that the paper was written in a hurry since it is very difficult to read. For example, what's after Line 96? It has not reached the bar of a scientific paper up to now. I suggest revising this paper in detail.

---
Reference
- Gong et al, Instance-Dependent Positive and Unlabeled Learning with Labeling Bias Estimation, TPAMI 2021.

**Questions:**

Please see "Weaknesses".

**Limitations:**

None.

---

> ### Author Rebuttal · Authors · 2023-08-10
>
> *Q1:* **The class prior is under estimated by using $n_P/n$ to replace $\pi$.**
>
> *A1:* Thanks for the nice conern. We actually use $N_P/n$ to replace $\pi$, that is, the proportion of the total number of positive samples to the total number of samples. $n_P=n_L$ is the number of labeled samples, $N_P$ is the number of positive samples, $n$ is the total number of samples, And $\sum_{j=1}^{n_P} \frac{1}{e(x_j^P)}=N_P$. Obviously, $N_P>n_P$. Therefore, the result is reasonable and does not underestimate the class prior.
>
> *Q2:* **The major problem of this paper is that it is very similar to [3]. It only changes the MLE method in [3] to a neural network.**
>
> *A2:* Thanks. Our work differs greatly from [3] with four aspects. First, the propensity score estimated in [3] is actually unidentifiable, and we stabilize our estimate of the propensity score by relaxing the Local Certainty assumption. Second, the deep learning model is used to estimate the propensity score instead of the linear model, which is better than logistic regression. Third, we improve the accuracy of cost-sensitive PU learning under biased labeling sets through normalized coupling commonly used PU learning, which is not involved in [3]. Finally, our method can be generalized to negative classes, and experiments are done (see PUbNe), which is not involved in [3]. The impact is great and the contribution is not small.
>
> Since our initial conditional assumptions are similar (based on the four basic assumptions of causal inference: Probabilistic, SUTVA, Consistent, unconfounded), the theoretical results look similar. In fact, our deep learning method coupling the cost-sensitive PU learning, which is different from [3].
>
> *Q3:* **This work actually works on “single-training-set PU learning”, while uPU and nnPU work on "case-control PU learning".**
>
> *A3:* Thanks for the nice conern. Our method can be implemented on both scenarios.
>
> The observed positive examples are generated from the same distribution in both scenarios. Hence, the learner has access to a set of examples drawn i.i.d. from the true distribution and a set of examples that are drawn from the positive distribution according to the labeling mechanism that is defined by the propensity score e(x). As a result, most methods（includes uPU and nnPU can handle both scenarios, but the derivation differs. Our method is coupled to cost-sensitive methods such as uPU, nnPU, and PUbN. Therefore, it can be implemented in two scenarios.
>
> In our paper, we mainly conduct experiments on the single-training-set, which is following the previous works.
>
> *Q4:* **My concern is that the model output is $p(s=1|x)$, instead of $p(s=1|x,y=1)$.**
>
> *A4:* Thanks for the nice concern. It's true that the estimate here is $p(s=1|x)$, but I'll show later that our estimate is reasonable, and that's what contributes to our relaxation of the Local Certainty assumption. In fact, we would point out that method in [3] of calculating the propensity score (PS) achieves this goal by maximizing the probability of observing the data, treating both the class posterior and the PS as latent variables. Unfortunately, this did not produce a identifiable PS; There are multiple false assumptions about the PS, which are as likely to be returned by this method as the true PS. Further, the estimated PS in the Local Certainty Scenario is something else, but in the Probabilistic Gap Scenario, the PS is unidentifable without additional assumptions. But in Local Certainty setting, we assume the relationship between the observed features and the true class is a deterministic function, this assumption is too strong to be realistic. We consider a large probabilistic gap. and positive instances that resemble negative instances are less likely to be labeled, i.e. $|P (y=1|x) -P(y=0|x)|>0.8$ and $P(y=1|x^L) > P(y=0|x^L)$, L means the labeled samples. According to the definition of probabilistic score, we have the $\sum e = n_p$ restriction, but $\hat{e}(x_i^N)>0$ on the negative will result in underestimated PS on labeled samples. Normalization can improve the estimation effect.
>
> Only labeled samples are considered. We have $0.9<P(y=1|x_i^L) \leq 1$. Obviously, $0.9e(x_i^L) < \hat{e}(x_i^L)\leq e(x_i^L)$. This shows that our estimated PS for the labeled sample is smaller than the true PS. Moreover, the PS of the estimated negative sample is greater than 0, i.e. $\hat{e}(x_i^N)>0$. According to the PS definition, we have the following equation $\sum \hat{e_i} = {n_p}$. In Local Certainty Scenario and [3], PS of negative samples also share part of the probability, the PS of the positive sample is underestimated. Therefore, we need to use the normalization method to improve the accuracy of our classification algorithm.
> $\frac{\frac{1}{e(x_i^L)}}{\sum_{j} \frac{1}{e(x_j^L)}} = A$.
>
> $0.9A < \frac{\frac{1}{\hat{e}(x_i^L)}}{\sum_j \frac{1}{\hat{e}(x_j^L)}} < 10A/9$. This shows that our estimate is relatively stable.
>
> *Q5:* **It seems that the paper was written in a hurry since it is very difficult to read. For example, what's after Line 96?**
>
> *A5:* Thanks for the advice. We will carefully check for spelling, grammar and typographical errors, thoroughly proofread the paper, improve the overall quality, credibility and readability of the paper, and carefully revise it in the final version.
> Line 96 should end with a period instead of a colon ":".

---

> > ### Author Response · Authors · 2023-08-12
> >
> > Dear Reviewer w1uD,
> >
> > We sincerely thank you for providing valuable feedback and suggestions.
> >
> > We have carefully considered your comments and have provided detailed responses addressing each of your concerns. We hope that our revisions have satisfactorily addressed your concerns.
> >
> > If you have any further questions or uncertainties regarding our work, please reach out to us. We would be more than happy to discuss them with you.
> >
> > Best,
> >
> > The Authors

---

> > ### Comment · Reviewer_w1uD · 2023-08-13
> > **Thanks for the rebuttal!**
> >
> > First, I thank the authors for the rebuttal.
> >
> > For Q1, my concerned is solved. I am sorry for my mistake. However, I still suggest the authors should *use visually distinguishable notations* for different symbols. Now it seems that it needs to know **the total number of P data in the unlabeled set**, which is a strong assumption. The authors should write it out clearly in the paper.
> >
> > For Q2, it has partly addressed my concern. From my point of view, the biggest difference is a NN. However, the output is biased (as I explained next).
> >
> > For Q3, my concern is resolved.
> >
> > For Q4, now we reach a consensus that the output is **biased**. However, the authors didn't discuss it explicitly in the paper. I think it is a big problem in the manuscript because the authors should rigorously point it out in the paper and then discuss the estimation error or assumptions. In the rebuttal, the authors say that they need several assumptions to reduce the bias. I did some derivations:
> > $$
> > p(s=1|x,y=1)=\frac{p(s=1,y=1|x)}{p(y=1|x)}=\frac{p(s=1|x)}{p(y=1|x)}.
> > $$
> > It means that only when the confidence of P data is very high and reaches 1, the equation holds. However, I do not think all the P data have an almost-one confidence. I think a more suitable method is to select high-confidence examples and then train the NN.
> >
> > Besides, I also had a question about the last equation in line 159. I did some derivations and got $E_{p(x|y=1)}e(x)=\frac{n_p}{N_p}$. It indicates that $\sum_{j=1}^{N_p}e(x)=n_p$, but I cannot get the equation in line 159. Could the authors elaborate it?
> >
> > Overall, since part of my question is solved. I will increase my score by 1.

---

> > > ### Author Response · Authors · 2023-08-14
> > > **Thanks for the reply!**
> > >
> > > *Q1:* **It needs to know the total number of P data in the unlabeled set, which is a strong assumption.**
> > >
> > > *A1:* In fact, in cost-sensitive PU learning, such as nnPU. This is a general assumption and has been omitted due to space constraints. For details, see Positive-unlabeled learning with non-negative risk estimator in neurips 2017. There is a lot of work on estimating $\pi$, such as the paper "Learning from corrupted binary labels via class-probability estimation" presented at ICML 2015, "Mixture proportion estimation via kernel embedding of distributions" presented at ICML 2016, "Estimating the class prior and posterior from noisy positives and unlabeled data" presented at NIPS 2016, " Class-prior estimation for learning from positive and unlabeled data" presented at Machine Learning 2017.
> > >
> > > *Q2:* **The biggest difference is a NN. However, the output is biased.**
> > >
> > > *A2:* Indeed, our biggest contribution is not only the use of NN, but also the use of regularization techniques to couple cost-sensitive PU learning to alleviate the general underestimation of propensity scores(PS). For details, see A4.
> > >
> > > *Q4-1:* **The output is biased. The authors should rigorously point it out in the paper and then discuss the estimation error or assumptions.**
> > >
> > > *A4-1:* Thank you very much for pointing out the question. Indeed, due to space limitations, it was not strictly pointed out in the paper that the PS estimates may be biased. We'll add this part to the final version of the paper and clarify it strictly. Another reason not stated is that in fact [3] this problem is not explicitly stated either. In fact, this problem can be understood from a simple point of view, because the unbiasedness needs to be guaranteed $\sum_{j=1}^{n}e(x_j)=n_p$, where $e(x_j^N)=0$, but actually because $\hat{e}(x_j^N)>0$, The value of $\hat{e}(x_j^P)$ will be partially apportioned. $\hat{e}(x_j^P)$ must be underestimated.
> > > In "Recovering the PS from Biased Positive Unlabeled Data" presented at AAAI 2022, this paper also points out that the PS estimated by [3] are not identifiable. So in addition to NN, another major contribution we made in this paper is to use regularization techniques to couple cost-sensitive PU learning to alleviate the general underestimation of PS. Instead of only using NN to estimate PS.
> > >
> > > *Q4-2:* **I do not think all the P data have an almost-one confidence. I think a more suitable method is to select high-confidence examples and then train the NN.**
> > >
> > > *A4-2:* Thank you for your advice. In fact, your "select high-confidence examples and then train the NN" is another method in PU learning, such as self-PU. We added experiments to illustrate this problem. It shows that our effect is better than self-PU on biased annotation sets.
> > >
> > > Supplemental Experiments on MINST. The setting conditions are the same as those in Table 1 of the supplementary materials.
> > >
> > > |method|Distripution|ACC|Prec|Rec|F1|AUC|AP|
> > > |-|-|-|-|-|-|-|-|
> > > |LRe | [.65,0,.15,0,.1,0,.07,0,.03,0] | 86.19(0.75) | 92.94(0.64) | 77.89(1.38) | 84.75(0.93) | 88.06(0.93) | 88.72(1.09)|
> > > | nnPUe | .. | 92.45 (1.61) | 90.45 (2.26) | 94.73 (1.24) | 92.53 (1.55) | 92.48 (1.60) | 88.29 (2.43)|
> > > |nnPU without normalize | .. | 90.95(1.61) | 88.18(2.40) | 94.38(2.74) | 91.13(1.56) | 91.00(1.61) |  85.98(2.25) |
> > > |Self-PU | .. | 90.08(0.47)  | 90.08 (0.47) | 89.35 (1.17) | 90.70 (1.73) | 90.00 (0.53) | 85.61 (0.69)|
> > >
> > > LRe: Logistic regression estimation of PS for PU learning.
> > >
> > > Well, it's really easy to explain, in fact, if the covariate x is almost certain of the data label y. That means all the P data has an almost-one confidence. However, if this condition is not met, the PN classification effect will be poor, and the upper limit of PU classification is PN classification. (All positive samples are marked, and all non-labeled samples are negative samples.) That is, it will definitely lead to worse PU classification. If you can't learn PN well, you can't learn PU well. Therefore, the problems solved by PU learning are problems with good PN classification effect. Otherwise, the problems discussed may be meaningless. Our method only requires high confidence of P data, which is in line with the actual application scenario of PU learning.

---

> > > > ### Author Response · Authors · 2023-08-14
> > > >
> > > > *Q5:* **I cannot get the equation in line 159. Could the authors elaborate it?**
> > > >
> > > > *A5:*  Sorry, there's no rigorous proof here, it's just discussed in line 212. We will supplement the discussion of this issue in the final version of the paper.
> > > >
> > > > First,  $n_P=n_L$ is the number of labeled samples, $N_P$ is the number of positive samples again. They're different.
> > > > $\sum_{j=1}^{N_p}e(x)=n_p$ and $\sum_{j=1}^{n_p}\frac{1}{e(x_j^P)}=N_p $ is established at the same time. The former sums all positive samples, while the latter sums only labeled samples.
> > > >
> > > > There are two explanations for line 159. One understanding is that, according to the PS definition, each labeled sample $x_j^P$ corresponds to $\frac{1}{e(x_j^P)}$ positive samples. So $\sum_{j=1}^{n_p}\frac{1}{e(x_j^P)}=N_p $ it's true. Because $P(x|s=1)= P(x, y=1|s=1)$, we have
> > > >
> > > > $E_{P(x|s=1)}\frac{1}{P(s=1|x,y=1)}$
> > > >
> > > > $=\sum P(x,y=1|s=1) \frac{1}{P(s=1|x,y=1)}$
> > > >
> > > > $=\sum \frac{P(s=1|x,y=1)P(x,y=1)}{P(s=1)} \frac{1}{P(s=1|x,y=1)}$
> > > >
> > > > $= \sum \frac{ P(x,y=1)} {P(s=1)}=\frac{n}{n_P}\sum P(x,y=1)$
> > > >
> > > > $= \frac{n}{n_P}\frac{N_P}{n}= \frac{N_P}{n_P}$.
> > > >
> > > > It indicates that $\sum_{j=1}^{n_p}\frac{1}{e(x_j^P)}=N_p $.

---

> > > > ### Comment · Reviewer_w1uD · 2023-08-14
> > > >
> > > > For Q1, I am sorry that I cannot agree with that it should be taken for granted to know the true number of positive data in PU learning literature. It is an *assumption*, and rigorously assumptions should be explicitly written out in papers. I have never seen any PU learning paper directing using *the number of true P data* but they all write it out in the paper.
> > > >
> > > > Besides, I think it is impossible to estimate the class prior here. Mixture proportion estimation is based on $p(x)=\pi p(x|y=1)+(1-\pi) p(x|y=-1)$ and having both data from $p(x)$ and $p(x|y=1)$. Here, we only have data from $p(x|y=1,s=1)$.
> > > >
> > > > For Q4-2,
> > > > >That means all the P data has an almost-one confidence.
> > > >
> > > > **I am sorry that I cannot agree with this point.** This point says that different classes are separated completely and do not have any overlap. It may not be true in many problems. Usually, we only have a subset points having $p(y=1|x)=1$. These points are the so-called **anchor points**, they are not all of the data set. You can refer to label-noise learning literature for more details, such as [1]. We need to select such data in the data set, instead of using all. Therefore, I think your method can be more valid by using the anchor points.
> > > >
> > > > [1] Classification with Noisy Labels by Importance Reweighting, TPAMI 2016.

---

> > > > > ### Author Response · Authors · 2023-08-17
> > > > >
> > > > > *Q1:* **It is an assumption, and rigorously assumptions should be explicitly written out in papers.**
> > > > >
> > > > > *A1:* Thank you for your suggestion. We will state this assumption that $\pi$ is assumed known in the final version. Specifically, we'll correct this in line 79: "We denote the class prior probability as $\pi = p(y = 1)$, where we follow the convention [1] to assume $\pi$ as known throughout the paper."
> > > > >
> > > > > *Q2:* **I think it is impossible to estimate the class prior.**
> > > > >
> > > > > *A2:* Thanks for the nice concern. Estimating the class prior is not the focus of our research. The main content of our paper is the improvement of cost-sensitive PU learning methods in the biased labeled set scenario. Most PU learning methods are established on a known class prior.
> > > > >
> > > > > Besides, there are some works which is able to estimate the class prior here with assumptions. For example, in [1], "We then extend the identifiability theory of class priors from the unbiased to the biased setting.". Or you can use the methods in [2] and [3] to estimate the probability of positive and negative samples in the bias case and use $\frac{\hat{N}_p}{n}$ to estimate $\pi$, where $\hat{N}_p$ is the total number of estimated positive samples.
> > > > >
> > > > > In our approach, you can adjust the hyperparameter $\pi$ to achieve better model results.
> > > > >
> > > > > *Q3:* **That means all the P data has an almost-one confidence.**
> > > > >
> > > > > *A3:* Sorry for the misleading statement. The proposed method does not require all the P data to have an almost-one confidence. When the positive and negative samples are completely inseparable, the question is almost unsolvable. In contrast, the proposed method works well under the conditions of relaxation of certain constraints.
> > > > >
> > > > > In fact, according to our formula, we only require a large confidence level for the labeled samples, because in fact, the propensity score only acts to modify the weight of the labeled samples. If the relationship between the observed features and the true class is probabilistic., there is a reasonable assumption that positive instances that resemble negative instances are less likely to be labeled. If $\beta <P(y=1|x_i^L) \leq 1$, then we have $\beta e(x_i^L) < \hat{e}(x_i^L)\leq e(x_i^L)$. Let's make $\frac{\frac{1}{e(x_i^L)}}{\sum_{j} \frac{1}{e(x_j^L)}} = A$. have $\beta A < \frac{\frac{1}{\hat{e}(x_i^L)}}{\sum_j \frac{1}{\hat{e}(x_j^L)}} < A /\beta$. actually when $\beta=0.9$ or even 0.8,0.7 is available. Experiments have shown that our method is greatly improved in this case. Therefore, the proposed method is usable.
> > > > >
> > > > >
> > > > > *Q4:* ** You can refer to label-noise learning literature for more details, such as [1]. We need to select such data in the data set, instead of using all. Therefore, I think your method can be more valid by using the anchor points.**
> > > > >
> > > > > *A4:* As mentioned in A3, we only modify the weight of the labeled samples. We do not modify the weight of all samples and there is no need to assume that anchor points exist. We only require a high degree of confidence in the labeled samples. In fact, there have been related papers on anchor points methods, see [3]. It assumes that $e=k*p(y=1|x)$ and anchor points exist. We have done comparative experiments to show that our method is more effective.
> > > > >
> > > > > Supplemental Experiments on MINST.
> > > > >
> > > > > | method  | labeled distripution | ACC(%) | Prec. (%) | Rec. (\%) | F1 (%) | AUC (%) | AP (%)|
> > > > > |-|-|-|-|-|-|-|-|
> > > > > | anchor | [0.65,0,0.15,0,0.10,0,0.07,0,0.03,0] | 88.22(0.95) | 94.66(1.41) | 80.70(3.15) | 87.06(1.37) | 92.36(1.92) | 93.37(2.33)|
> > > > > | nnPUe | [0.65,0,0.15,0,0.10,0,0.07,0,0.03,0] | 92.45 (1.61) | 90.45 (2.26) | 94.73 (1.24) | 92.53 (1.55) | 92.48 (1.60) | 88.29 (2.43)|
> > > > >
> > > > > Anchor:The method in Probabilistic Gap Propensity Estimation of [3].
> > > > >
> > > > > [1] Class Prior Estimation with Biased Positives and Unlabeled Examples, AAAI 2020.
> > > > >
> > > > > [2] Beyond the selected completely at random assumption for learning from positive and unlabeled data, Springer, 2020.
> > > > >
> > > > > [3] Recovering the Propensity Score from Biased Positive Unlabeled Data, AAAI 2022

---

> > > > > > ### Comment · Reviewer_w1uD · 2023-08-17
> > > > > > **Thanks for your reply!**
> > > > > >
> > > > > > First, I thank authors for their time and effort in the rebuttal.
> > > > > >
> > > > > > Maybe my concern is not expressed clearly. My concern is not your formulation in in Eq.(6). I think all the content before line 182 is right. My concern lies in "Estimating the Propensity Scores from Data". It seems that it needs **all the selected positive data to be anchor points**, according to my derivation. If we use the authors' assumption that $\beta\leq p(y=1|x_{i}^{L}) \leq 1$, the problem still exist. When we train a binary classifier with selected positive data sampled from $p(x|s=1)$ and unlabeled data sampled from $p(x|s=0)$, the output is the approximation of $p(s=1|x)$ biased from $p(s=1|x,y=1)$. I think there exists estimation error in this part. Therefore, I still think that it is problematic.

---

> > > > > > > ### Author Response · Authors · 2023-08-18
> > > > > > >
> > > > > > > Thank you very much for your reply.
> > > > > > >
> > > > > > > First, we admit that there exists estimation bias when the P data confidence is low, as we said in A4-2. " It was not strictly pointed out in the paper that the PS estimates may be biased. We will add this part to the final version of the paper and clarify it strictly. " However, as we stated in our previous reply, it is clear that to be unbiased, we have to make $E_{p(x|y=1)}=\frac{n_p}{N_p}$, which indicates that $\sum_{j=1}^{N_p}e(x)=n_p$. As A4-1 indicates, since $e(x_i^N)>0$. Existing propensity score estimation methods such as SAR-EM in [1] often underestimate the propensity score of positive samples, which will cause bias or produce an unidentifiable result (overfitting) that all labeled samples have a propensity score close to 1 and all unlabeled samples have a propensity score close to 0. For example, under the Probabilistic Gap condition, [2] assume that there are anchor points, and also assume that $[P(y=1|x)]^2=\frac{P(s=1|x)}{K}$. This actually means that the distribution of $P(y=1|x)$ and the ideal propensity score are known.
> > > > > > >
> > > > > > > These methods either produce deviations in the same direction, overfitting is not available or unidentifiable errors, or actually assume that the distribution of $P(y=1|x)$ and the ideal propensity score are known. And these methods are usually linear, which is likely not practical. This makes it difficult to use these methods in practice and to obtain the desired results.
> > > > > > >
> > > > > > > And there is a theory that the propensity score is unrecognizable without additional assumptions, see [2]. It is inevitable that estimation bias exists in this part. This leads us to develop an algorithm that produces a range of acceptable deviations and ultimately works better under relaxed assumptions, unknown $P(y=1|x)$ distributions and ideal propensity scores. Finally, we choose to use NN to estimate the propensity score, and use regularization to alleviate the co-directional deviation problem. Coupling the cost-sensitive PU learning method to improve the performance of the cost-sensitive PU learning method under the biased labeling set. and the effect is better than that of the existing methods.
> > > > > > >
> > > > > > > [1] Beyond the selected completely at random assumption for learning from positive and unlabeled data, Springer, 2020.
> > > > > > >
> > > > > > > [2] Recovering the Propensity Score from Biased Positive Unlabeled Data, AAAI 2022

---

> > > > > > > ### Author Response · Authors · 2023-08-20
> > > > > > >
> > > > > > > Thanks for your valuable questions and feedback during the rebuttal phase of the review process. We have tried our best to addressed your concerns and provided detailed responses.
> > > > > > >
> > > > > > > As the rebuttal phase is nearing its conclusion, we kindly request your prompt response regarding our rebuttal, as it would greatly assist us in finalizing the manuscript. Thanks again for your nice suggestions.

---

> > > > > > > > ### Comment · Reviewer_w1uD · 2023-08-20
> > > > > > > >
> > > > > > > > Thanks for your explanations that previous methods all have bias or assumptions about the propensity score.
> > > > > > > >
> > > > > > > > However, it is quite difficult to figure out which method has a lower bias. Better classification performance does not indicate lower bias of the estimation of the propensity score.
> > > > > > > >
> > > > > > > > **The current version of the paper just regards all the P data as anchor points without any discussions or explanations about the estimation error of the propensity score.** I think it is a very big problem from a rigorous perspective although the mistake is quite concealing.
> > > > > > > >
> > > > > > > > I think by considering anchor points for the estimation, e.g. selecting anchor points from the P data can strengthen the work. I think more discussion and better estimation methods can improve the paper.

---

> > > > > > > > > ### Author Response · Authors · 2023-08-20
> > > > > > > > >
> > > > > > > > > Thanks for your nice concern. We can indeed consider anchors for estimation, such as selecting anchors from P data, and other better estimation methods can improve the paper. However, we would like to kindly state that the current approach has already been an effective way to perform right now.
> > > > > > > > >
> > > > > > > > > In this case, our method proposes corresponding solutions for the above three questions: co-directional deviation, overfitting, and known propensity score. The normalization technique is used to make the deviations no longer in the same direction (co-directional error will produce worse results). Moreover, the maximum estimation error ratio of sample weights is reduced. The regularization technique is used to alleviate the overfitting problem, and the work does not assume a known propensity score or a positive sample distribution.
> > > > > > > > >
> > > > > > > > > Co-direction error produces worse results: $P(\hat{Y}|E, L)=\frac{1}{n} \sum_{i=1}^n s_i(\frac{1}{e_i}\delta_1+(1-\frac{1}{e_i})\delta_0)+(1-s_i) \delta_0$ (Formula 1) underestimates the propensity score of s=1 labeling samples, resulting in increases the cost of $\delta_1$ and decreases the cost of $\delta_0$ on the labeled samples.
> > > > > > > > >
> > > > > > > > >  After normalization is used, $\theta A < \frac{\frac{1}{\hat{e}(x_i^L)}}{\sum_j \frac{1}{\hat{e}(x_j^L)}} < A /\theta$.$\frac{\frac{1}{e(x_i^L)}}{\sum_{j} \frac{1}{e(x_j^L)}} = A$.$\theta <P(y=1|x_i^L) \leq 1$ does not have such co-direction error.
> > > > > > > > >
> > > > > > > > > We may assume that the error of propensity scores estimated by the NN method is the same as that estimated by the linear method. (In fact, the NN methods are usually more general, which may produce results with less error.) That is, the estimate of the propensity score has a maximum error ratio of $\beta$, with $\beta e(x_i^L) \leq \hat{e}(x_i^L)\leq e(x_i^L)$. of the following shows that our regularization technique can yield a smaller error ratio with respect to sample weights. Obviously, the sample $x_i^L$ has a sample weight of $\frac{1}{n\hat{e}(x_i^L)}$. in (Formula 1) with an error bound of $bias(\frac{1}{n\hat{e}(x_i^L)}) \leq \frac{1}{ne(x_i^L)}(\frac{1}{\beta}-1)$.
> > > > > > > > >
> > > > > > > > > In our approach, Sample $x_i^L$ has a weight of $\pi \frac{\frac{1}{\hat{e}(x_i^L)}}{\sum_{j} \frac{1}{\hat{e}(x_j^L)}}$.$P(\gamma e(x_i^L) < \hat{e}(x_i^L)\leq e(x_i^L))= \alpha $ where the set of samples is $S_1$ .$P(\beta e(x_i^L) < \hat{e}(x_i^L)\leq \gamma e(x_i^L))= 1-\alpha $ where the set of samples is $S_2$.$\beta<\gamma<1$ and $\sum_{i \in S_1}\frac{1}{e(x_i^L)} = \sum_{i \in S_2}\frac{1}{e(x_i^L)} = B$. So that $\sum_{j} \frac{1}{e(x_j^L)} = 2B =N_p $. For $x_i^L \in S_1$, we have $\frac{1}{e(x_i^L)} \leq \frac{1}{\hat{e}(x_i^L)} < \frac{1}{\gamma e(x_i^L)}$. For $x_i^L \in S_2$, we have $\frac{1}{\gamma e(x_i^L)} \leq \frac{1}{\hat{e}(x_i^L)} < \frac{1}{\beta e(x_i^L)}$, so we can get $B(1+\frac{1}{\gamma}) \leq \sum_{j} \frac{1}{\hat{e}(x_j^L)} < B(\frac{1}{\gamma} + \frac{1}{\beta}) $ and we have $bias(\pi \frac{\frac{1}{\hat{e}(x_i^L)}}{\sum_{j} \frac{1}{\hat{e}(x_j^L)}}) \leq $$\max [\frac{1}{ne(x_i^L)}(\frac{2}{(1+\gamma)}-1), \frac{1}{ne(x_i^L)}(1-\frac{2}{\frac{1}{\beta}+\frac{1}{\gamma}}), \frac{1}{ne(x_i^L)}(\frac{2}{(1+\frac{1}{\gamma})\beta}-1), \frac{1}{ne(x_i^L)}(1-\frac{2}{\frac{\gamma}{\beta}+1})] <\frac{1}{ne(x_i^L)}(\frac{1}{\beta}-1)$ and obviously we have $\frac{2}{(1+\gamma)} <\frac{2}{(1+\frac{1}{\gamma})\beta} < \frac{1}{\beta}, 0<1-\frac{2}{\frac{\gamma}{\beta}+1}<1-\frac{2}{\frac{1}{\beta}+\frac{1}{\gamma}} < 1-\beta < \frac{1}{\beta}-1, $ which shows that our regularization technique has a smaller error ratio with respect to sample weights.
> > > > > > > > >
> > > > > > > > > Thanks again for the nice suggestions. We will include the discussion in the final version.

---

> > > > > > > > > > ### Comment · Reviewer_w1uD · 2023-08-21
> > > > > > > > > >
> > > > > > > > > > Dear authors,
> > > > > > > > > >
> > > > > > > > > > Thanks for the response. Yes, adding the regularization and normalization technique can reduce the estimation error.
> > > > > > > > > >
> > > > > > > > > > However, the estimated value is still $p(s=1|x)$. If we directly use it for P data as $p(s=1|y=1, x)$, we need $p(y=1|x)=1$. It means all the P data should be anchor points. I am still concerned about this problem and I wonder what the estimation error is because it is very hard to satisfy the condition.

---

> > > > > > > > > > > ### Author Response · Authors · 2023-08-21
> > > > > > > > > > >
> > > > > > > > > > > Sorry for the unclear statement. The estimation error can be derived as below.
> > > > > > > > > > >
> > > > > > > > > > > First of all, we just need to focus on the labeled samples, because we only modified the weight of the labeled samples, instead of the weight of the unlabeled samples. We assume that we have $\theta <P(y=1|x_i^L) \leq 1$. The assumption is simple. And we have $p(s=1|y=1, x_i^L)=\frac{p(s=1, y=1|x_i^L)}{p(y=1|x_i^L)}= \frac{p(s=1|x_i^L)}{p(y=1|x_i^L)}$.
> > > > > > > > > > >
> > > > > > > > > > > We make $\frac{\frac{1}{e(x_i^L)}}{\sum_{j} \frac{1}{e(x_j^L)}} = A_i$.
> > > > > > > > > > > We can get $\theta A_i < \frac{\frac{1}{\hat{e}(x_i^L)}}{\sum_j \frac{1}{\hat{e}(x_j^L)}} < A_i /\theta$, which is the margin of the estimate of $A_i$. It depends on $\theta$. Obviously, as you say, when $\theta = 1$, this estimate is unbiased.  **Although this condition is difficult to satisfy, we can relax this assumption to estimate its error.**
> > > > > > > > > > >
> > > > > > > > > > > When $\theta \geq 0.9$, the estimated range is between $0.9A_i$ and $10A_i/9$. The estimated error is smaller than $A_i/9$, which is not a big error.

---

> > > > > > > > > > > > ### Comment · Reviewer_w1uD · 2023-08-21
> > > > > > > > > > > >
> > > > > > > > > > > > Thanks for the reply. Yes, if we make such assumptions, it will make the proposed method more valid. Since some of my concern has been addressed, I am willing to increase my score.
> > > > > > > > > > > >
> > > > > > > > > > > > However, as I said before, many key assumptions are missing in the submission version.
> > > > > > > > > > > > First, the absence of the assumption "**known number of positive data in unlabeled data**" results in my misunderstanding of the formulation. Second, **the assumption you have just discussed** is very critical since the method is wrong without such assumptions. Besides, the writing of the paper is not good with too many typos. These reasons result in my overall rating.

---

> > > > > > > > > > > > > ### Author Response · Authors · 2023-08-21
> > > > > > > > > > > > >
> > > > > > > > > > > > > Dear Reviewer,
> > > > > > > > > > > > >
> > > > > > > > > > > > > Thank you for your response. We appreciate your willingness to increase your score and your acknowledgment of the addressed concerns.
> > > > > > > > > > > > >
> > > > > > > > > > > > > We apologize for the missing key assumptions in the submission version, particularly the absence of the assumption regarding the known number of positive data in unlabeled data. We understand that this omission led to a misunderstanding of the formulation, and we apologize for any confusion caused. We will explicitly state in the paper "known number of positive data in unlabeled data". In line 79: "We denote the class prior probability as $\pi=p(y=1)$, where we follow the convention to assume $\pi$ as known throughout the paper. And we can actually estimate in the case of biased labeled samples $\pi$.
> > > > > > > > > > > > >
> > > > > > > > > > > > > Additionally, we acknowledge the critical nature of the assumption we discussed in our previous response and its impact on the validity of the proposed method. We will ensure that these assumptions are explicitly stated and properly emphasized in the revised manuscript. We will specify what assumptions we need between line 115-116. The disadvantages of existing methods, and the advantages of our methods. And the direction that may be improved in the future. We will include the following statements
> > > > > > > > > > > > >
> > > > > > > > > > > > > " Progress has been made in PU learning under biased labeling data, such as Beyond the selected completely at random assumption for learning from positive and unlabeled data, Springer, 2020 and Recovering the Propensity Score from Biased Positive Unlabeled Data, AAAI 2022. They all use $P(\hat{Y}|E, L)=\frac{1}{n} \sum_{i=1}^n s_i(\frac{1}{e_i}\delta_1+(1-\frac{1}{e_i})\delta_0)+(1-s_i) \delta_0$ (Formula 1) to train the model.
> > > > > > > > > > > > >
> > > > > > > > > > > > > However, it is clear that to be unbiased, we have to make $E_{p(x|y=1)}=\frac{n_p}{N_p}$, which indicates that $\sum_{j=1}^{N_p}e(x)=n_p$. But $\hat{e}(x_i^N)>0$, so that propensity score estimation methods such as SAR-EM in [1] often underestimate the propensity score of positive samples, which will cause bias or produce an unidentifiable result (overfitting) that all labeled samples have a propensity score close to 1 while all unlabeled samples have a propensity score close to 0. Or in case of the Probabilistic Gap condition, [2] assume that there are anchor points, and also assume that $[P(y=1|x)]^2=\frac{P(s=1|x)}{K}$. This actually means that the distribution of $P(y=1|x)$ and the ideal propensity score are known. And these methods are usually linear, which is likely not practical. This makes it difficult to use these methods in practice and to obtain the desired results. And there is a theory that the propensity score is unrecognizable without the assumption $p(y=1|x^L)=1$, see [2]. It is inevitable that estimation bias exists in this part. This leads us to develop an algorithm with NN that produces a range of acceptable deviations and ultimately works better under relaxed assumptions, unknown $P(y=1|x)$ distributions and ideal propensity scores. Our assumption is $\theta <P(y=1|x_i^L) \leq 1$, and the closer $\theta $ is to 1, the better our approach works.
> > > > > > > > > > > > >
> > > > > > > > > > > > > Finally, we choose to use NN to estimate the propensity score, prevent overfitting with regularization techniques, and use regularization to alleviate the co-directional deviation problem. Coupling the cost-sensitive PU learning method to improve the performance of the cost-sensitive PU learning method under the biased labeling set. and the effect is better than that of the existing methods.”
> > > > > > > > > > > > >
> > > > > > > > > > > > > We will then put the detailed formulas and supplementary experiments discussed above into the supplementary material to complete the overall description of our paper.
> > > > > > > > > > > > >
> > > > > > > > > > > > >
> > > > > > > > > > > > > Furthermore, we apologize for the issues with the writing style, including the presence of numerous typos. We recognize the importance of clear and concise writing in effectively presenting our ideas. In the revised version, we will dedicate significant effort to improving the writing quality and eliminating any remaining errors.
> > > > > > > > > > > > >
> > > > > > > > > > > > > We genuinely appreciate your valuable feedback, which has provided us with valuable insights for enhancing our work. We assure you that we will make every effort to address these concerns and improve the overall quality of the paper.
> > > > > > > > > > > > >
> > > > > > > > > > > > > Thank you once again for your time and thoughtful evaluation of our work.
> > > > > > > > > > > > >
> > > > > > > > > > > > > Best regards,
> > > > > > > > > > > > >
> > > > > > > > > > > > > Authors

---

### Official Review · Reviewer_fhZG · 2023-07-03

**Soundness:** 3 good
**Presentation:** 3 good
**Contribution:** 3 good
**Rating:** 7
**Confidence:** 4

**Summary:**

This paper considers PU learning issue. Existing cost-sensitive-based methods often rely on strong assumptions that examples with
an observed positive label were selected entirely at random. This work relaxes the assumption and proposes a unbiased novel method PUe based on the causal theory.

**Strengths:**

1. The motivation to relax the Selected Completely At Random assumption is practical and interesting.

2. Generalization theory to guarantee the effectiveness.

3. Excellent experiment performance to guarantee the effectiveness of the algorithm.

**Weaknesses:**

I am very confused on how to estimate the prior pi.

Please conduct experiments with wild OOD detection method Training OOD Detectors in their Natural Habitats.

I have not got where you have used the causal theory. Could you point out the causual theory you have used and clarify what causal assumptions you have used in you theorems?

**Questions:**

See weaknesses

---

> ### Author Rebuttal · Authors · 2023-08-10
>
> *Q1:* **I am very confused on how to estimate the prior $\pi$.**
>
> *A1:* Thanks. There is a lot of work on estimating $\pi$, such as the paper "Learning from corrupted binary labels via class-probability estimation" presented at ICML 2015, "Mixture proportion estimation via kernel embedding of distributions" presented at ICML 2016, "Estimating the class prior and posterior from noisy positives and unlabeled data" presented at NIPS 2016, " Class-prior estimation for learning from positive and unlabeled data" presented at Machine Learning 2017.
>
> Common methods for estimating $\pi$ include kernel method, kernel logistic regression, mixture proportion estimation and univariate transforms.
>
> Estimates on $\pi$ are not our main contribution. In papers on PU learning loss functions, $\pi$ is usually assumed to be known. Our main contribution is four points. Firstly, we relax the assumption of local certainty and propose a simple method to estimate the propensity score under the assumption of SAR, and apply it to PU learning to improve the performance of the algorithm. Second, using deep learning to estimate propensity score (PS) rather than linearity is better than logistic regression. Third, our PS estimation method can be extended to the case of negative classes (e.g. PUbNe). Fourth, our new algorithm can simply be coupled with most cost-sensitive algorithms to improve performance, as shown in the paper and supplementary experiments. We will include more analysis and results in the final version.
>
> *Q2:* **Please conduct experiments with wild OOD detection method "Training OOD Detectors in their Natural Habitats".**
>
> *A2:* Thank you very much for your question. Wild OOD detection can also be regarded as a PU problem. $P_{wild}=\pi P_{out}+(1-\pi)P_{in}$.
> We separate in-distribution (ID) data and OOD data from input data by using a small number of labeled samples and a large number of wild data samples. We can regard ID data as P class, OOD data as N class, wild data as unlabeled data, and labeled ID data as label data.
> Wild OOD detection can be tested and compared with our algorithm. However, the common dataset of PU work in the previous literature is not this OOD task, so I didn't consider doing this experiment at first. And due to time constraints and additional experiments, we did not complete code refactoring and experimental simulations. We will supplement this experimental comparison in the final paper to further explore the performance of our algorithm.
>
> *Q3:* **Could you point out the causal theory you have used and clarify what causal assumptions you have used in you theorems?**
>
> *A3:* Thanks for your question. There are four basic assumptions in causal inference. 1st, assignment mechanism must be probabilistic, $0<P(s=1|x_i, y_i=1) \leq1$. for positive sample, (It can be extended to negative classes, see PUbNe) If the selection probability is 0, the sample is difficult to accurately determine without additional assumptions. 2nd, SUTVA means that the potential outcome (label value) of any individual does not change depending on whether other individuals are treated or not. 3rd. Consistent, there is no noise. 4th, the core assumption on which our work relies, is that the assignment mechanism must be unconfounded, requiring that all the assignment probabilities $P(s |x_i, t_i=1)$ are free from dependence on the potential outcomes. allocation mechanism is independent of potential outcomes.
>
> And we think that this allocation mechanism should be completely dependent on the covariate that can be observed in the sample, i.e. $e(x_i)$. Usually, PU learning studies the case of $e(x_i)=c$. Here we look at the case where $e(x_i)$ is a variable function, that is, the SCR hypothesis (which does not conform to the SCAR hypothesis). And under the unconfounded hypothesis, according to the theory of causal inference, we can know
> $s_i \perp x_i |e(x_i)$, or, equivalently, $e(x_i)=P(s_i=1|x_i, e(x_i),y_i=1)= P(s_i=1|e(x_i),y_i=1)$. and $s_i \perp y_i(0),y_i(1) |e(x_i)$. where $y_i(0), y_i(1)$ is the potential result, and in PU learning there is $y_i(0)=y_i(1)=1$, i.e., the final potential result is positive regardless of whether the positive class is selected into the annotation set.
>
> We are actually using causal reasoning backwards here, first completing the causal analysis of the processing (i.e. selection) and the label itself, and then using known causal analysis, combining selected and unselected samples to determine the distribution of positive samples.
>
> Determine the causality: In PU case, accepting processing means selecting samples for labeling. The result refers to whether the sample itself is a positive sample. Whether a sample is a positive sample is determined when the sample is generated, and the behavior of selecting a sample for marking is executed only after the sample is generated.
> In causal inference, what happens after the result is produced has no effect on the result, i.e. whether the sample is selected or not has a causal effect on the label value is 0. In essence, individual causality is also 0, which ensures that we can estimate the positive probability of samples based on selected samples and unselected samples.
>
> In causal inference, we can think that the samples in each block are generated by completely randomized experiments, and estimate the causal effects in each block separately. Given knowledge of the propensity score, the sample weights can also be adjusted using propensity score adjustment weights to estimate causal effects. The individual causal effect is 0, and we know the distribution of covariates of the labeled and unlabeled samples, so we can estimate the positive probability with the estimated propensity score. Simply put, we want to increase the weight of samples with low labeling probability and reduce the weight of samples with high labeling probability. What kind of weight adjustment is used is based on the inverse probability weighting in causal inference.

---

### Official Review · Reviewer_7yJd · 2023-07-05

**Soundness:** 3 good
**Presentation:** 1 poor
**Contribution:** 2 fair
**Rating:** 4
**Confidence:** 4

**Summary:**

This paper considers the problem that existing cost-sensitive-based methods often rely on strong assumptions that examples with an observed positive label were selected entirely at random in Positive-Unlabeled (PU) learning. The authors propose a PU learning enhancement (PUe) algorithm based on causal inference theory, which employs normalized propensity scores and normalized inverse probability weighting (NIPW) techniques to reconstruct the loss function, thus obtaining a consistent, unbiased estimate of the classifier and enhancing the model’s performance. Moreover, they investigate and propose a method for estimating propensity scores in deep learning using regularization techniques when the labeling mechanism is unknown. Experiments on three benchmark datasets demonstrate the proposed PUe algorithm significantly improves the accuracy of classifiers on non-uniform label distribution datasets compared to advanced cost-sensitive PU methods.

**Strengths:**

1. One strength of this paper is its focus on addressing the prevalent issue of bias in the labeled set, which often does not conform to the Selected Completely At Random (SCAR) assumption. This assumption posits that the observed labeled examples are a random subset of the complete set of positive examples. In many real-world scenarios, this assumption does not hold true, leading to biased learning outcomes.

**Weaknesses:**

1.  This paper contains several spelling errors, which can potentially hinder the reader's understanding and overall perception of the work. For instance, on line 131, the term "oftreament" appears to be a typographical error where two words are mistakenly joined. Similarly, on line 149, "porpensity" is likely a misspelling of "propensity," and on line 189, "Apparemtly" should be corrected to "Apparently." Lastly, on line 220, "pi" might be a typo or a symbol that is not clearly defined in the context. These errors suggest a lack of thorough proofreading and can detract from the overall quality and credibility of the paper. It is highly recommended that the authors conduct a careful review of the manuscript for spelling, grammar, and typographical errors before final submission.
2. A significant weakness of this paper lies in its lack of clear explanation and connection between the problem it aims to solve - the bias in the labeled set not conforming to the Selected Completely At Random (SCAR) assumption - and the causal inference method it employs to address this issue. Specifically, lines 123 to 145 appear to attempt to elucidate this connection, but the explanation falls short due to the lack of clear definitions and elaboration on key concepts. This lack of clarity can make it difficult for readers to understand the rationale behind the proposed method and its relevance to the problem.
3. A notable weakness in the experimental section of the paper is the presentation of results in Tables 1 and 2. The current format does not clearly highlight the best-performing algorithm, making it difficult for readers to quickly and intuitively discern which method is superior and by what margin.

**Questions:**

Could the authors provide a more detailed explanation of how the causal inference method addresses the issue of bias in the labeled set not conforming to the SCAR assumption?

**Limitations:**

 A significant limitation of this paper is its overall clarity and readability.

---

> ### Author Rebuttal · Authors · 2023-08-10
>
> *Q1:* **This paper contains several spelling errors.**
>
> *A1:* Thank you for your advice. We will carefully check spelling, grammar and typographical errors, thoroughly proofread the paper, improve the overall quality and credibility of the paper, and carefully revise it in the final version. Lines 131, 149, 189 do have errors. In line 220, "pi" should actually be "$\pi$".
>
> *Q2:* **Could the authors provide a more detailed explanation of how the causal inference method addresses the issue of bias in the labeled set not conforming to the SCAR assumption?**
>
> *A2:* Thanks. There are four basic assumptions in causal inference. First, assignment mechanism must be probabilistic, $0<P(s=1|x_i, y_i=1) \leq1$. for positive sample, (It can be extended to negative classes, see PUbNe) If the selection probability is 0, the sample is difficult to accurately determine without additional assumptions. Second, SUTVA means that the potential outcome (label value) of any individual does not change depending on whether other individuals are treated or not. Third. Consistent, there is no noise. Fourth, the core assumption on which our work relies, is that the assignment mechanism must be unconfounded, requiring that all the assignment probabilities $P(s |x_i, t_i=1)$ are free from dependence on the potential outcomes. allocation mechanism is independent of potential outcomes.
>
> And we think that this allocation mechanism should be completely dependent on the covariate that can be observed in the sample, i.e. $e(x_i)$. Usually, PU learning studies the case of $e(x_i)=c$. Here we look at the case where $e(x_i)$ is a variable function, that is, the SCR hypothesis (which does not conform to the SCAR hypothesis). And under the unconfounded hypothesis, according to the theory of causal inference, we can know
> $s_i \perp x_i |e(x_i)$, or, equivalently, $e(x_i)=P(s_i=1|x_i, e(x_i),y_i=1)= P(s_i=1|e(x_i),y_i=1)$. and $s_i \perp y_i(0),y_i(1) |e(x_i)$. where $y_i(0), y_i(1)$ is the potential result, and in PU learning there is $y_i(0)=y_i(1)=1$, i.e., the final potential result is positive regardless of whether the positive class is selected into the annotation set.
>
> We are actually using causal reasoning backwards here, first completing the causal analysis of the processing (i.e. selection) and the label itself, and then using known causal analysis, combining selected and unselected samples to determine the distribution of positive samples.
>
> Determine the causality: In PU case, accepting processing means selecting samples for labeling. The result refers to whether the sample itself is a positive sample. Whether a sample is a positive sample is determined when the sample is generated, and the behavior of selecting a sample for marking is executed only after the sample is generated.
> In causal inference, what happens after the result is produced has no effect on the result, i.e. whether the sample is selected or not has a causal effect on the label value is 0. In essence, individual causality is also 0, which ensures that we can estimate the positive probability of samples based on selected samples and unselected samples.
>
> In causal inference, we can think that the samples in each block are generated by completely randomized experiments, and estimate the causal effects in each block separately. Given knowledge of the propensity score, the sample weights can also be adjusted using propensity score adjustment weights to estimate causal effects. The individual causal effect is 0, and we know the distribution of covariates of the labeled and unlabeled samples, so we can estimate the positive probability with the estimated propensity score. Simply put, we want to increase the weight of samples with low labeling probability and reduce the weight of samples with high labeling probability. What kind of weight adjustment is used is based on the inverse probability weighting in causal inference.
>
> According to Rubin's formulas for Weighting Estimators and Blocking Estimators:
>
> $\hat{\tau}^{ht}= \frac{1}{N}\sum_{i=1}^{N} \frac{s_{i}{Y_{i}}^{P}}{e({x}^{P}_{i})} $
>
> $-\frac{1}{N} {\sum_{i=1}}^N \frac{(1-s_i){Y_i}^{P}}{1-e({x_i}^{P})}$
>
> $\hat{\tau}^{dif}(j)=\frac{\sum_{i:B_{i}(j)=1}s_i {Y_{i}}^{P}}{\sum_{i:B_{i}(j)=1}s_i}- \frac{\sum_{i:B_{i}(j)=1}(1-s_i) {Y_{i}}^{P}}{\sum_{i:B_{i}(j)=1}(1-s_i)}$
>
> Only positive are labeled, so $Y^{P}_{i}=1$. Because the causal effect is 0. We have the following equation:
>
> $\frac{1}{N_{p_j}}\sum_{i=1}^{N_{p_j}} \frac{s_{i}Y^{P}_{i}}{e({x_i}^{P})} $
>
> $= \frac{1}{N_{p_j}}\sum_{i=1}^{N_{p_j}} \frac{(1-s_i){Y_i}^{P}}{1-e({x_i}^{P})}$,
>
> $N_{p_j}$ is the number of positive samples in the jth block.
>
> The jth block represents a positive sample with a propensity score between $b_j$ and $b_j+\epsilon$,
>
>  $b_j \in [0,1)$, $\epsilon > 0$ and $b_j+\epsilon \leq 1$. $s_i=1$ means sample i is selected into the labeled set.
>
> So we can get $T_j= \sum_{i=1}^{n_{p_j}} \frac{1}{e({x_i}^{L})}=\sum_{i=1}^{N_{p_j}-n_{p_j}} \frac{1}{1-e({x}^{'}_{i})}$,
>
> ${n_{p_j}}={n_{L_j}}$ is the number of labeled samples in the jth block, ${x}^{'}_{i}$ are unlabeled positive samples.
>
> $(\frac{b_j}{b_j+\epsilon})N_{p_j} \leq \frac{n_{p_j}}{b_j+\epsilon} < T_j \leq \frac{n_{p_j}}{b_j} < (\frac{b_j+\epsilon}{b_j})N_{p_j}$.
>
> Make $\epsilon \to 0$, $T_j \to N_{p_j}$. So $\sum T_j = \sum_{i=1}^{n_{L}} \frac{1}{e(x^{L}_{i})} = N_p$, which is used in the unbiased proof in the paper.
>
> *Q3:* **A notable weakness in the experimental section of the paper is the presentation of results in Tables 1 and 2. The current format does not clearly highlight the best-performing algorithm.**
>
> *A3:* Thanks for your advice, we will mark the best results, clearly highlighting the best performing algorithms so that readers can quickly identify them. In the final version of the paper, we will refine the presentation of the results.

---

> > ### Author Response · Authors · 2023-08-12
> >
> > Dear reviewer 7yJd:
> >
> > We sincerely thank you for the review and comments.
> >
> > We have provided corresponding responses and results, including a more detailed explanation of how the causal inference method addresses the issue, which we've tried our best to cover your concerns.
> >
> > Please let us know whether your concerns have been well addressed. We would like to further discuss with you if you still have any unclear parts of our work.
> >
> > Best,
> >
> > The Authors

---

> > > ### Comment · Reviewer_7yJd · 2023-08-14
> > >
> > > Thank you for your prompt response; it has addressed most of my concerns. However, there are still a few points that I would like to further explore for clarity.
> > >
> > > You mentioned that "the sample weights can also be adjusted using propensity score adjustment weights to estimate causal effects." Could you elucidate how the sample weights and propensity scores are connected? An example illustrating this relationship would be particularly helpful.
> > >
> > > You also noted, "What kind of weight adjustment is used is based on the inverse probability weighting in causal inference." I'm curious to understand the rationale behind using the inverse form of the propensity scores for sample weight adjustments. Why is the reciprocal approach chosen in this context?

---

> > > > ### Author Response · Authors · 2023-08-17
> > > >
> > > > *Q1:* **Could you elucidate how the sample weights and propensity scores are connected? An example illustrating this relationship would be particularly helpful.**
> > > >
> > > > *A1:* Let's take a simple example in causal inference (although the propensity score is discrete). We want to test the average treatment effect of drug A on people with disease B $\tau$. Assume that the male-to-female ratio of disease B in the population is 1:1. We selected 100 male and 100 female patients, 70 male and 40 female patients were treated with drug A, and the remaining patients were treated with placebo as a control group. Here, the propensity score is 0.7 for men and 0.4 for women, S=1 for treatment with drug A, S=0 for placebo, 1 to 100 for men, 101 to 200 for women, $s_i = 1$ indicates that person i was treated with drug A, $s_i = 0$ indicates that person i was treated with placebo, and Y indicates the effect of drug A or placebo.
> > > >
> > > > Obviously, it is not reasonable to use $E[Y|S=1]- E[Y|S=0]$ directly to estimate $\tau=E[Y(1)]- E[Y(0)]=E[E(Y|X, S=1)]- E[E(Y|X, S=0)]$.
> > > > $\hat{\tau_1}=\frac{1}{110}\sum_{i=1}{200} s_iy_i - \frac{1}{90}\sum_{i=1}{200} (1-s_i)y_i $
> > > >
> > > > Instead, the inverse probability weighted $E[\frac{YS}{e(X)}]- E[\frac{YS}{e(X)}]$ of the propensity score should be used to estimate $\tau=E[E(Y|X, S=1)]- E[E(Y|X, S=0)]$.
> > > >
> > > > $\hat{\tau_2}=\frac{1}{200}\sum_{i=1}{100} \frac{s_iy_i }{0.7}+\frac{1}{200}\sum_{i=101}{200} \frac{s_iy_i }{0.4}- \frac{1}{200}\sum_{i=1}{100} \frac{(1-s_i)y_i }{1-0.7}+\frac{1}{200}\sum_{i=101}{200} \frac{(1-s_i)y_i }{1-0.4} $.
> > > >
> > > > When his PS is continuous, sample weights and propensity scores have the same relationship. Here's a brief description of the theory. Under unconfoundedness, $S\bot Y(0), Y(1)|X$, Two properties $S\bot X|e(X)$ and $S\bot Y(0), Y(1)|e(X)$ can be deduced from $e(X)=P(S=1|X)$. $\tau=E[E(Y|X, S=1)]- E[E(Y|X, S=0)]= E[E(Y|e(X),S=1)]- E[E(Y|e(X),S=0)]$. can be obtained based on the properties (Rosenbaum and Rubin suggest that they be divided into five layers) and the average causal effect estimation can be obtained. The successive versions of the stratification are the following weighted estimates:
> > > > $\tau= E[Y(1)]- E[Y(0)]= E[\frac{SY}{e(X)}]- E[\frac{(1-S)Y}{1-e(X)}]$
> > > >
> > > >
> > > > *Q2:* **Rationale behind using the inverse form of the propensity scores for sample weight adjustments. Why is the reciprocal approach chosen in this context?**
> > > >
> > > > *A2:* The basis for adjusting sample weights using the propensity score inverse probability form is shown in A1.
> > > >
> > > > $\tau= E[Y(1)]- E[Y(0)]= E[\frac{SY}{e(X)}]- E[\frac{(1-S)Y}{1-e(X)}]$
> > > >
> > > > There are four common methods for causal inference: Regression Estimators, Weighting Estimators by Propensity Score Inverse Probabilities, Blocking Estimators That Use the Propensity Score, and Matching Estimators.
> > > >
> > > > We have considered all these methods. Regression Estimators, especially linear models, are the most studied. However, we think that linear models have limited expressive ability, especially in the CV field, are more likely to be nonlinear models. Supplementary comparative tests were also made to illustrate this problem. Weighted estimation by propensity score inverse probabilities is the method of choice, and we think it's a natural combination of cost-sensitive PU learning. The training instances are properly reweighted.
> > > >
> > > > Supplemental Experiments on MINST. The setting conditions are the same as those in Table 1 of the supplementary materials.
> > > >
> > > > |method|Distripution|ACC|Prec|Rec|F1|AUC|AP|
> > > > |-|-|-|-|-|-|-|-|
> > > > |LRe | [0.65,0,0.15,0,0.10,0,0.07,0,0.03,0] | 86.19(0.75) | 92.94(0.64) | 77.89(1.38) | 84.75(0.93) | 88.06(0.93) | 88.72(1.09)|
> > > > | nnPUe | [0.65,0,0.15,0,0.10,0,0.07,0,0.03,0] | 92.45 (1.61) | 90.45 (2.26) | 94.73 (1.24) | 92.53 (1.55) | 92.48 (1.60) | 88.29 (2.43)|
> > > >
> > > > LRe: Logistic regression estimation of propensity scores for PU learning.
> > > >
> > > > The Blocking Estimators That Use the Propensity Score method is considered at the beginning, but it is difficult to determine the number of layers. When the layers are small enough, the method is equivalent to the Weighted estimation by propensity score inverse probabilities method. Matching Estimators is also a method that can be considered. However, it is very difficult to find an exact match when there are many covariates, requiring a distance metric. However, we have not fully considered this method. Therefore, this method is not used currently.
> > > >
> > > > However, one direction that could be considered for future work is to use multiple approaches to Mixed Estimators. For example, first layering, and then using a matching method within each block to estimate the sample classification.

---

> > > > > ### Comment · Reviewer_7yJd · 2023-08-17
> > > > >
> > > > > Dear Authors,
> > > > >
> > > > > Thank you for your detailed response, which has addressed many of my concerns. I recognize and appreciate the efforts you have made in addressing my concerns. This paper offers elegant theoretical guarantees, and through their approach, one can achieve a tighter generalization bound. This is something I deeply appreciate.
> > > > >
> > > > > However, I feel compelled to point out that the paper appears to have been written in haste. There are numerous typos throughout the document, and certain sections are verbose and somewhat convoluted, making them challenging to follow. This is especially evident in Section 2.2, where the relationship of causal inference and Pu learning is discussed. The clarity and quality of writing are essential for conveying complex ideas to readers, and as it stands, this paper requires substantial revision in that regard.
> > > > >
> > > > > Given the above, while I have decided to slightly raise my score, I still retain my reservations about the paper.
> > > > >
> > > > > Best regards

---

> > > > > > ### Author Response · Authors · 2023-08-18
> > > > > >
> > > > > > Dear Reviewer,
> > > > > >
> > > > > > We would like to express our gratitude for your thorough feedback and valuable insights. Your comments have been instrumental in guiding us towards improving our manuscript. We wholeheartedly acknowledge the efforts we need to invest in addressing the concerns you have raised.
> > > > > >
> > > > > > We are pleased to hear that you appreciate the elegant theoretical guarantees and the potential for achieving a tighter generalization bound offered by our approach. It encourages us to further refine our work.
> > > > > >
> > > > > > We apologize for the presence of numerous typos and the verbosity observed in certain sections of the paper. We fully recognize the importance of clarity and conciseness in effectively conveying our ideas to readers. Specifically, we acknowledge that Section 2.2, which discusses the relationship between causal inference and Pu learning, requires better organization and explanation. We will diligently revise the manuscript to rectify these issues and ensure a smoother and more coherent flow of information.
> > > > > >
> > > > > > We will recap the definitions of SCAR and SAR between lines 101 and 122 to reduce the length and complexity of the chapter.
> > > > > > We will modify the description of the relationship between causal inference and PU learning between lines 123 to 151, clarify the four basic assumptions, and then explain the relationship between causal inference and PU learning. That is, reverse causal inference is used to determine that the individual causal effect is 0 and is used to infer the distribution of positive samples. Then provide a more detailed explanation of how the causal inference method addresses the issue of bias in the labeled set not conforming to the SCAR assumption. Then we further explain the common methods of causal inference and why we chose inverse probability weighting as the basis for adjusting sample weights.
> > > > > > Finally, we will carefully check the full text and correct the clerical errors in the original paper. For example, on line 96, ": " should be replaced by ".". On line 131, the term "oftreament" should be "of treatment". On line 149, "porpensity" should be "propensity", on line 189, "Apparemtly" should be "Apparently.", and on line 220, "pi" should be "$\pi$". And we will mark the best results, clearly highlighting the best performing algorithms so that readers can quickly identify them.
> > > > > >
> > > > > > We genuinely appreciate your decision to slightly raise the score, indicating that our efforts have been recognized. However, we understand that your reservations about the paper still persist, and we are committed to addressing them comprehensively.
> > > > > >
> > > > > > Once again, we sincerely thank you for your time and thoughtful evaluation of our work. Your feedback is invaluable, and we will use it to enhance the clarity, quality, and overall presentation of our manuscript.
> > > > > >
> > > > > > Best regards,
> > > > > > Authors

---

> > > > > > ### Author Response · Authors · 2023-08-20
> > > > > >
> > > > > > Thanks for your valuable questions and feedback during the rebuttal phase of the review process. We have tried our best to addressed your concerns and provided detailed responses.
> > > > > >
> > > > > > As the rebuttal phase is nearing its conclusion, we kindly request your prompt response regarding our rebuttal, as it would greatly assist us in finalizing the manuscript. Thanks again for your nice suggestions.

---

### Official Review · Reviewer_nitz · 2023-07-09

**Soundness:** 2 fair
**Presentation:** 3 good
**Contribution:** 2 fair
**Rating:** 6
**Confidence:** 3

**Summary:**

The paper introduces an algorithm for Positive-Unlabeled (PU) learning that tackles the problem of biased labeled data. By utilizing causal inference theory, the algorithm utilizes normalized propensity scores and inverse probability weighting to reconstruct the loss function and achieve an unbiased estimate of the classifier. Compared to existing cost-sensitive PU methods, the proposed PUe algorithm demonstrates significant improvements in classifier accuracy, particularly on datasets with non-uniform label distributions. Additionally, the paper proposes a method for estimating propensity scores in deep learning models when the labeling mechanism is unknown. Overall, the PUe framework effectively addresses selection bias, enhances PU learning, and improves model performance.

-------------------------------
After rebuttal: I would like to increase my rate a little bit. However, I think the novelty part still needs more clarification.

After the author-review discussion: I could increase my rate a little bit for the thorough rebuttal provided by the authors. However, I think the rebuttal provides a lot of information that may not be easily included in the paper. I would like to see a plan on how to incorporate the reviews/rebuttals into the paper.

During the final discussion: The plan has given me more confidence in how the paper can be if we accept it.

**Strengths:**

Importance of the studied problem (biased positive data): The problem of studying biased positive data is highly significant and practical. In many real-world scenarios, the selection of positive samples is often biased, while existing methods typically assume that the selection of positive samples is completely random. The proposed PUe algorithm in this paper addresses this issue and effectively handles biased labeled data, filling a research gap in the field of PU learning.

Well-written: The paper is well-written, with clear descriptions of the problem background, method principles, and experimental design and results. The authors introduce causal inference theory and utilize techniques such as normalized propensity scores and inverse probability weighting to provide a viable solution, which is presented in a clear and concise manner.

Experimental results demonstrate the effectiveness of the method: The paper provides experimental evidence of the proposed PUe algorithm's effectiveness.

**Weaknesses:**

Lack of novelty: One of the limitations of the paper is that it may lack novelty. Similar work, such as the paper "Recovering the Propensity Score from Biased Positive Unlabeled Data" presented at AAAI 2022, may have explored similar ideas and methods earlier. It is important for a research contribution to demonstrate novelty and differentiate itself from existing approaches in the field.

Limited causal inference aspect: While the paper is motivated by causal inference, it falls short in incorporating substantial causal elements. The approach of reweighting samples based on sampling probabilities is straightforward and may not fully capture the complexity of causal relationships. To strengthen the causal inference aspect, the paper could have explored additional methods or frameworks that explicitly address causal assumptions and mechanisms.

Lack of significant contribution: The primary contribution of the paper should lie in the proposed method for estimating propensity scores. However, the approach of using model predictions as weights for propensity scores is not fundamentally different from classical self-supervised methods. Although a regularizer is introduced, it may not provide a meaningful improvement. Additionally, the process of normalizing the estimated propensity scores could have been done separately after learning the scores, making the proposed regularizer less essential or impactful.

**Questions:**

Comparison with "Recovering the Propensity Score from Biased Positive Unlabeled Data" presented at AAAI 2022: A direct comparison between the paper and the mentioned work would require a thorough analysis of both papers.

Significance of causal inference in the context of the proposed work: Although the paper mentions causal inference as a motivation, it is valid to question the significance of causal inference in this work. If the process of generating biased data is explicitly defined and the proposed loss function can be straightforwardly derived from it, the direct application of causal inference may not be evident. It is important to consider whether the paper effectively incorporates causal inference methods or frameworks that address causal assumptions and mechanisms beyond straightforward reweighting.

Comparison with self-supervised methods like selfPU: Evaluating the performance of the proposed approach compared to self-supervised methods like selfPU would require a detailed analysis and comparison of the methodologies and experimental results from both approaches.

**Limitations:**

Yes

---

> ### Author Rebuttal · Authors · 2023-08-08
>
> *Q1:* **Comparison with paper in AAAI 2022.**
>
> *A1:* Thanks for the nice concern. The main content of aaai's article is introduced and explained respectively in the case of Local Certainty and Probabilistic Gap Scenario PS estimation which is s identifiable. There are four points about the novelty and contribution of this paper. First, the assumption of Local Certainty is relaxed, and the algorithm effect is improved. Second, use deep learning to estimate propensity scores(PS), rather than linearity, better than logistic regression. Third, our PS estimation method can be extended to the case with negative classes (e.g., PUbNe). Fourthly, our new algorithm can be simply coupled with most cost-sensitive algorithms to improve performance, as shown in paper. We’ll include more analysis and results in final version.
>
> 1. The paper relaxes the assumption of the Local Certainty and does not require the relationship between the observed features and the true class is a deterministic function . When x has a large Probabilistic Gap to y and positive instances that resemble negative instances are less likely to be labeled, this algorithm can also be true, i.e. $|P (y=1|x) -P(y=0|x)|>0.8$ and  $P(y=1|x^L) > P(y=0|x^L)$,  L means the labeled samples.
>
> And the Local Certainty method in the aaai paper is used to estimate the PS, the $\sum e = n_p$ restriction and $\hat{e}(x_i^N)>0$ on the negative sample will result in underestimated PS on labeled samples. Normalization can improve the estimation effect.
>
> Only labeled samples are considered. We have $0.9<P(y=1|x_i^L) \leq 1$. Obviously, $0.9e(x_i^L) < \hat{e}(x_i^L)\leq e(x_i^L)$. This shows that our estimated PS for the labeled sample is smaller than the true PS. Moreover, the PS of the estimated negative sample is greater than 0, i.e. $\hat{e}(x_i^N)>0$. According to the PS definition, we have the following equation $\sum \hat{e_i} = {n_p}$. In aaai Local Certainty Scenario, PS of negative samples also share part of the probability, the PS of the positive sample is underestimated. Therefore, we need to use the normalization method to improve the accuracy of our classification algorithm.
> $\frac{\frac{1}{e(x_i^L)}}{\sum_{j} \frac{1}{e(x_j^L)}} = A$.
>
> $0.9A < \frac{\frac{1}{\hat{e}(x_i^L)}}{\sum_j \frac{1}{\hat{e}(x_j^L)}} < 10A/9$. This shows that our estimate is relatively stable.
>
> 2. In the Probabilistic Gap, We don't limit the PS to be linear. Deep learning is used for estimation. The regularization method is used to prevent overfitting, and the normalization method is used to alleviate the problem of underestimating PS. We added experiments on MINST using LR (Logistic regression) to estimate PS, showing that LR is inferior to deep learning.
>
> |Method|Distribution|ACC|Prec|Rec|F1|AUC|AP|
> |-|-|-|-|-|-|-|-|
> |LR | [.65,0,.15,0,.1,0,.07,0,.03,0] | 86.19(0.75) | 92.94(0.64) | 77.89(1.38) | 84.75(0.93) | 88.06(0.93) | 88.72(1.09)|
> | nnPUe | .. | 92.45 (1.61) | 90.45 (2.26) | 94.73 (1.24) | 92.53 (1.55) | 92.48 (1.60) | 88.29 (2.43)|
> |nnPUe w/o normalize | .. | 90.95(1.61) | 88.18(2.40) | 94.38(2.74) | 91.13(1.56) | 91.00(1.61) |  85.98(2.25) |
> |Self-PU | .. | 90.08(0.47)  | 90.08 (0.47) | 89.35 (1.17) | 90.70 (1.73) | 90.00 (0.53) | 85.61 (0.69)|
>
> *Q2:* **Significance of causal inference in the context of the proposed work.**
>
> *A2:* Thanks for the nice question. The simple method is applied because we analyze the causality here is simple. Our main contributions are indeed not here. See the first question for our main contributions. We'll explore more complex ways to study this in the future.
>
> We're using causal reasoning in reverse here. In general, we study the average causal effect of a treatment. However, we first complete the causal analysis of the processing (i.e. selection) and the label itself, and then use the known causal analysis, combined with the selected samples and the unselected samples to determine the distribution of positive samples.
>
> In this paper, accept processing means that the sample is selected for labeled. And the result is refers to whether the sample itself is a positive sample. Without noise in the detection, whether the sample itself is a positive sample is determined at the time when the sample is generated, and the behavior of selecting is performed after the sample is generated. In causal inference, what happens after the result is produced has no effect on the result, i.e. whether the sample is selected has a causal effect on the label value is 0. In essence, individual causality is also 0.
>
> Based on such a simple causality, we use the method of reweighting the sample to estimate the positive probability of the sample.
>
> *Q3:* **The primary contribution of the paper should lie in the proposed method for estimating PS.**
>
> *A3:* It makes sense for us to introduce a regularizer here. The regularization method is used to solve the overfitting problem, and the normalization method is used to solve the problem of small estimated PS, which is a method to deal with different problems.
>
> Ablation experiment (in the above table) showed that when the regularizer was removed, the performance was reduced. This will make it very easy to overfit, when we estimate the PS. The estimated PS in the label set is close to 1, with unlabeled is close to 0. The regularization method smooths the estimated PS. The regularization method is to solve the problem that the model underestimates PS. We also do additional experiments here to demonstrate that the performance of the model deteriorates when the regularization effect is removed under the nnPUe algorithm.
>
> *Q4:* **Comparison with selfPU.**
>
> *A4:* According to the aaai paper, it cannot estimate identifiable PS without making certain assumptions about the data. But according to the formula we gave in the first question, it's approximate. This is not explained by the self-monitoring method. Results in the above table show that our scheme is better than self-PU in the case of biased label datasets.

---

> > ### Author Response · Authors · 2023-08-12
> >
> > Dear reviewer nitz:
> >
> > We sincerely thank you for the review and comments.
> >
> > We have provided corresponding responses and results, which we've tried our best to cover your concerns. With extra experiments during the rebuttal period, the advantage of our approach becomes clearer. Some missing comparison have been provided and will be added to the final paper.
> >
> > Please let us know whether your concerns have been well addressed. We would like to further discuss with you if you still have any unclear parts of our work.
> >
> > Best,
> >
> > The Authors

---

> > ### Comment · Reviewer_nitz · 2023-08-13
> > **Re**
> >
> > Thank you for your thorough response addressing my concerns. I appreciate your efforts to clarify several points. However, I would like to seek more clarification or details on novelty before I am confident to change my score.
> >
> > In light of my concern about novelty, could you provide additional analysis or evidence that clearly distinguishes your method from existing approaches, highlighting the aspects that set your work apart? Specifically, while your response sheds light on the novelty of your contributions compared to the AAAI 2022 paper, I would appreciate more specific examples illustrating how your relaxation of the Local Certainty assumption, deep learning-based PS estimation, extension to negative classes, and coupling with cost-sensitive algorithms uniquely differentiate your approach from existing methods.
> >
> > Regards,

---

> > > ### Author Response · Authors · 2023-08-15
> > >
> > > *Q1&2:* **Examples of relaxation of the Local Certainty assumption and deep learning-based PS estimation**
> > >
> > > *A1&2:* Here we merge the two issues since they are relevant . According to the AAAI 2022 paper: In the Local Certainty scenario, we assume the relationship between the observed features and the true class is a deterministic function $f:X\rightarrow Y$, where X is the feature space and Y = {0, 1}, while allowing the propensity score to be an arbitrary probability function. but this assumption is too strong. Actually follow our reply. Only $|P (y=1|x) -P(y=0|x)|>0.8$ and $P(y=1|x^{L}) > P(y=0|x^{L})$, needs to be satisfied using our method to use the estimated propensity score well for sample classification. (where L represents the labeled sample and is used to distinguish it from the positive class.)
> > >
> > > An obvious example is that in MNIST, the accuracy of existing algorithms is more than 90%, but cannot reach 100%. So it's not allowing the propensity score to be an arbitrary function, it's supposed to be a linear function. However, according to our experimental results, the probability function fitted using deep learning is better than the linear estimation. 6-layer MLP is used for mnist datasets, 13-layer CNN is used for cifar-10, and ResNet-50 is used for Alzheimer datasets.
> > >
> > > Supplemental Experiments on MINST. The setting conditions are the same as those in Table 1 of the supplementary materials.
> > >
> > > | method  | Distripution | ACC(%) | Prec. (%) | Rec. (\%) | F1 (%) | AUC (%) | AP (%)|
> > > |-|-|-|-|-|-|-|-|
> > > |LRe | [0.65,0,0.15,0,0.10,0,0.07,0,0.03,0] | 86.19(0.75) | 92.94(0.64) | 77.89(1.38) | 84.75(0.93) | 88.06(0.93) | 88.72(1.09)|
> > > | nnPUe | [0.65,0,0.15,0,0.10,0,0.07,0,0.03,0] | 92.45 (1.61) | 90.45 (2.26) | 94.73 (1.24) | 92.53 (1.55) | 92.48 (1.60) | 88.29 (2.43)|
> > >
> > > LRe: Logistic regression estimation of propensity scores for PU learning.
> > >
> > > Here's our theoretical explanation of this example.
> > >
> > > $e(x)=P(s=1|x,y=1)=\frac{P(x|s=1)P(s=1)}{P(y=1|x)P(x)}$
> > >
> > > When $|P (y=1|x) -P(y=0|x)|>0.8$. Only labeled samples are considered. We have $0.9<P(y=1|{x_i}^{L}) \leq 1$. Obviously, $0.9e({x_{i}}^{L}) < \frac{P(s=1)P({x_i}^L|s=1)}{P({x_i}^L)}=\hat{e}({x_{i}}^{L})\leq e({x_i}^{L})$. This shows that our estimated propensity score for the labeled sample is smaller than the true propensity score. Moreover, the propensity score of the estimated negative sample is greater than 0, i.e. $\hat{e}({x_{i}}^{N})>0$. And according to the propensity score definition, we have the following equation $\sum \hat{e_i} = {n_p}$. In aaai Local Certainty Scenario, propensity scores of negative samples also share part of the probability, the propensity score of the positive sample is underestimated. Therefore, we need to use the normalization method to improve the accuracy of our classification algorithm. $\frac{0.9\frac{1}{e({x_{i}}^{L})}}{\sum_{j} \frac{1}{e({x_{j}}^{L})}} < \frac{\frac{1}{\hat{e}({x_{i}}^{L})}}{\sum_{j} \frac{1}{\hat{e}({x_{j}}^{L})}} <\frac{\frac{1}{e({x_{i}}^{L})}}{0.9\sum_{j} \frac{1}{e({x_{j}}^{L})}}$. This shows that our estimate is relatively stable.

---

> > > > ### Author Response · Authors · 2023-08-15
> > > >
> > > > *Q3:* **Examples of extension to negative classes.**
> > > >
> > > > *A3:* In this paper, we consider the case where negative classes are also selectively labeled, and do related experiments (see PUbN), which is not mentioned in the AAAI 2022 paper. We allow the propensity score of negative samples to be greater than 0 in the case of negative samples with selective labeling.
> > > >
> > > > Let $\pi = p(y=+1)$ and $\rho = p(y=-1,s=+1)$. Both $\pi$ and $\rho$ are hyperparameters.
> > > >
> > > > The PUbN formula is as follows:
> > > >
> > > > Let $\sigma(x)=p(s=+1|x)$, however, the $\sigma(x)$ is actually unknown,we should replace $\sigma(x)$ by its estimate $\hat{\sigma}(x)$.We can get the classification risk of PUbN ($R_{PUbN}(g)$), as the following expression:
> > > >
> > > > $R_{PUbN}(g)=\pi R_P(g,+1)+\rho R_{bN}(g,-1)+\bar{R}_{s=-1,\eta,\hat{\sigma}}(g),$
> > > >
> > > > where $\bar{R}_{s=-1,\eta,\hat{\sigma}}(g) = $
> > > >
> > > > $E_{x\sim p(x)}[\mathbb{1}_{\hat{\sigma}(x)\leq \eta}L(-g(x))(1-\hat{\sigma}(x))]$
> > > >
> > > > $+\pi E_{x\sim p_P(x)}[\mathbb{1}_{\hat{\sigma}(x) > \eta}L(-g(x))\frac{1-\hat{\sigma}(x)}{\hat{\sigma}(x)}]$
> > > >
> > > > $+\rho E_{x\sim p_{bN}(x)}[\mathbb{1}_{\hat{\sigma}(x) > \eta}L(-g(x))\frac{1-\hat{\sigma}(x)}{\hat{\sigma}(x)}].$
> > > >
> > > > Then $R_{bN}(g,-1)$ and $\bar{R}_{s=-1,\eta,\hat{\sigma}}(g)$ can also be approximated from data by
> > > >
> > > > $\hat{R}_{bN}(g,-1) =$
> > > >
> > > > $\frac{1}{n_{bN}} \sum_{i=1}^{n_{bN}}L(g(x^{bN}_{i}), -1).$
> > > >
> > > > $\hat{\bar{R}}_{s=-1,\eta,\hat{\sigma}}(g)$
> > > >
> > > > $=\frac{1}{n_U}\sum^{n_U}_{i=1}$
> > > >
> > > > $[\mathbb{1}_{\hat{\sigma}(x^U_i)\leq \eta} L(g(x^U_i), -1)(1-\hat{\sigma}(x^U_i))]$
> > > >
> > > > $+\frac{\pi}{n_P}\sum^{n_P}_{i=1}$
> > > >
> > > > $[\mathbb{1}_{\hat{\sigma}(x^P_i) > \eta} L(g(x^P_i), -1)\frac{1-\hat{\sigma}(x^P_i)}{\hat{\sigma}(x^P_i)}]$
> > > >
> > > > $+\frac{\rho}{n_bN}\sum^{n_{bN}}_{i=1}$
> > > >
> > > > $[\mathbb{1}_{\hat{\sigma}(x^{bN}_i) > \eta} L(g(x^{bN}_i), -1)\frac{1-\hat{\sigma}(x^{bN}_i)}{\hat{\sigma}(x^{bN}_i)}].$
> > > >
> > > > We can get the empirical loss of PUbN:
> > > >
> > > > $\hat{R}_{PUbN,\eta,\hat{\sigma}}(g)=\pi \hat{R}_P(g,+1)$
> > > >
> > > > $+\rho\hat{R}_{bN}(g,-1)$
> > > >
> > > > $+\hat{\bar{R}}_{s=-1,\eta,\hat{\sigma}}(g).$
> > > >
> > > > Our PUbNe empirical loss is as follows:
> > > >
> > > > $\hat{R}_{PUbN \hat{e}, \eta,\hat{\sigma}}(g)=\pi \hat{R}^{\hat{e}}_P(g,+1)$
> > > >
> > > > $+\rho\hat{R}^{\hat{e}}_{bN}(g,-1)$
> > > >
> > > > $+\hat{\bar{R}}^{\hat{e}}_{s=-1,\eta,\hat{\sigma}}(g),$
> > > >
> > > > where $\hat{R}_{bN}^{\hat{e}}(g,-1)$
> > > >
> > > > $= \sum_{i=1}^{n_{bN}}\frac{1}{\tilde{\hat{e}}(x_i^{bN})}L(g(x_i^{bN}), -1).$
> > > >
> > > > Here, $\frac{1}{\tilde{\hat{e}}(x_i^{bN})}$ is reciprocal normalization of estimated PS of bN samples.
> > > >
> > > > And $\hat{\bar{R}}^{\hat{e}}_{s=-1,\eta,\hat{\sigma}}(g)$
> > > >
> > > > $=\frac{1}{n_U}\sum^{n_U}_{i=1}$
> > > >
> > > > $[\mathbb{1}_{\hat{\sigma}(x^U_i)\leq \eta} L(g(x^U_i), -1)(1-\hat{\sigma}(x^U_i))]$
> > > >
> > > > $+\pi\sum^{n_P}_{i=1}$
> > > >
> > > > $[\frac{1}{\tilde{\hat{e}}(x_i^{P})}\mathbb{1}_{\hat{\sigma}(x^P_i) > \eta} L(g(x^P_i), -1)\frac{1-\hat{\sigma}(x^P_i)}{\hat{\sigma}(x^P_i)}]$
> > > >
> > > > $+\rho\sum^{n_{bN}}_{i=1}$
> > > >
> > > > $[\frac{1}{\tilde{\hat{e}}(x_i^{bN})}\mathbb{1}_{\hat{\sigma}(x^{bN}_i) > \eta} L(g(x^{bN}_i), -1)\frac{1-\hat{\sigma}(x^{bN}_i)}{\hat{\sigma}(x^{bN}_i)}].$
> > > >
> > > >
> > > > The experimental results are as follows:
> > > >
> > > > Supplemental Experiments on MINST. The setting conditions are the same as those in Table 1 of the supplementary materials.
> > > >
> > > > | Dataset | method | Distripution | ACC(%) | Prec. (%) | Rec. (\%) | F1 (%) | AUC (%) | AP (%)|
> > > > |-|-|-|-|-|-|-|-|-|
> > > > |MNIST |PUbN | [0.65,0,0.15,0,0.10,0,0.07,0,0.03,0] | 87.07 (1.73) | 83.43 (2.74) | 92.21 (2.55) | 87.55 (1.55) | 87.14 (2.74) | 89.73 (2.34)|
> > > > |MNIST| PUbNe | [0.65,0,0.15,0,0.10,0,0.07,0,0.03,0] | 94.70 (0.63) | 97.16 (1.59) | 91.33 (1.58) | 94.46 (0.71) | 94.66 (0.50) | 94.66 (0.64)|
> > > > |CIFAR-10 |PUbN | [0.72,0.15,0,0,0,0,0,0,0.1,0.03] | 84.54 (0.54) | 82.18 (1.38) | 78.40 (2.70) | 80.20 (0.99) | 91.62 (0.27) | 87.31 (0.40)|
> > > > |CIFAR-10| PUbNe | [0.72,0.15,0,0,0,0,0,0,0.1,0.03] | 85.66 (0.58) | 83.18 (1.63) | 80.52 (2.29) | 81.79 (0.66) | 92.69 (0.74) | 89.41 (1.24)|

---

> > > > > ### Author Response · Authors · 2023-08-15
> > > > >
> > > > > *Q4:* **Examples of coupling with cost-sensitive algorithms.**
> > > > >
> > > > > *A4:* According to the aaai 2022 paper: Using Propensity Sores For Classification. $P(\hat{Y}|E,L)=\frac{1}{n} \sum_{i=1}^n s_i(\frac{1}{e_i}\delta_1+(1-\frac{1}{e_i})\delta_0)+(1-s_i) \delta_0$（Formula 1）. However, according to the identifiable propensity score property, $\sum_{j=1}^{n}e(x_j)=n_p$ needs to be guaranteed, where $e(x_j^N)=0$. However, in fact, the value of $\hat{e}(x_j^P)$ will be partially apportioned due to $\hat{e}(x_j^N)>0$. This must make $\hat{e}(x_j^P)$ underestimated. The aaai paper uses Formula 1 to train downstream classifiers for each data set to achieve class posterior estimation, which increases the cost of $\delta_1$  and decreases the cost of $\delta_0$ on the labeled samples. This makes the propensity score for classification estimated directly using the aaai 2022 paper worse.
> > > > >
> > > > > Then we found that in cost-sensitive PU learning technique, the training instances are properly reweighted. and a common way to score propensity is inverse probability weighting (a reweighting technique). Conditionally, the two methods can be combined to solve the defects of the two methods themselves. The disadvantage of the cost-sensitive PU learning technique is that the classification effect of the application deteriorates when the labeled samples are biased. The propensity score is estimated by the fact that the propensity score of positive samples is always underestimated, which leads to the decline of the classification effect of the model. This is why we chose to couple with cost-sensitive PU learning.

---

> > > > > > ### Author Response · Authors · 2023-08-15
> > > > > > **Thanks for the reply**
> > > > > >
> > > > > > Dear Reviewer,
> > > > > >
> > > > > > Thank you for your inquiries. In response to your questions, we have provided additional analysis or evidence that clearly distinguishes our method from existing approaches. We have specifically highlighted the aspects that make our work stand out.
> > > > > >
> > > > > > We kindly request you to review our responses and assess whether they effectively address your concerns. If you have any further questions or require more clarification, we welcome continued discussion on the matter.
> > > > > >
> > > > > > Thank you for your attention and engagement.
> > > > > >
> > > > > > Best regards,
> > > > > > Authors

---

> > > > > > > ### Comment · Reviewer_nitz · 2023-08-16
> > > > > > > **Re**
> > > > > > >
> > > > > > > Thank you for your response.
> > > > > > >
> > > > > > > I am satisfied with your response and would like to increase my score a little bit (from 4 to 5). However, I think you may need to provide a plan on how to update your paper to include the reviews into consideration. I would stick to the score 5 unless there is a clear plan for updating the paper. Thanks.
> > > > > > >
> > > > > > > Regards,

---

> > > > > > > > ### Author Response · Authors · 2023-08-18
> > > > > > > >
> > > > > > > > Thanks for the nice suggestion. Below is the plan for updating the paper.
> > > > > > > >
> > > > > > > > We will add the main content of the paper that mentions AAAI 2022 between line 47 and line 48. And explain the limitations of AAAI 2022 paper and our improvement and contribution to it. After line 68, explain the novelty and contribution of our work. Supplemental experiments were added to the supplemental materials.
> > > > > > > >
> > > > > > > > We will explain Significance of causal inference in the context of our work on line 118. And add a description of role of normalization techniques on line 160. Supplemental experiments were added to the supplemental materials.
> > > > > > > >
> > > > > > > > The comparative experiments with self-PU will be added to the supplementary materials. The results of our method and self-PU experiments are analyzed and compared.
> > > > > > > >
> > > > > > > > We are going to give examples of relaxation of the Local Certainty assumption and deep learning-based PS estimation between 199 line and 200 line.
> > > > > > > >
> > > > > > > > We will give examples of extension to negative classes after 231line, and put the formula of PUbN and PUbNe in the supplementary material.
> > > > > > > >
> > > > > > > > We will give examples of coupling with cost-sensitive algorithms between line 217 and line 225 to illustrate the reasons and benefits of coupling propensity scores with cost-sensitive PU learning methods.
> > > > > > > >
> > > > > > > > Finally, we will carefully check the full text and correct the clerical errors in the original paper. For example, on line 131, the term "oftreament" should be "of treatment". On line 149, "porpensity" should be "propensity", on line 189, "Apparemtly" should be "Apparently.", and on line 220, "pi" should be "$\pi$".
> > > > > > > >
> > > > > > > > Regards,
> > > > > > > > Authors

---

> > > > > > > > ### Author Response · Authors · 2023-08-20
> > > > > > > >
> > > > > > > > Thanks for your valuable questions and feedback during the rebuttal phase of the review process. We have tried our best to addressed your concerns and provided detailed responses.
> > > > > > > >
> > > > > > > > As the rebuttal phase is nearing its conclusion, we kindly request your prompt response regarding our rebuttal, as it would greatly assist us in finalizing the manuscript. Thanks again for your nice suggestions.

---

### Official Review · Reviewer_U7Dp · 2023-07-11

**Soundness:** 3 good
**Presentation:** 2 fair
**Contribution:** 3 good
**Rating:** 7
**Confidence:** 4

**Summary:**

This paper noted that existing Positive-Unlabeled (PU) learning methods assumed that positive samples are selected entirely at random, ignoring the prevalent selection bias in real-world PU problems. To overcome this limitation, the authors proposed a PU learning enhancement (PUe) algorithm based on causal inference theory. They adapted sample re-weighting methods, to be specific, inverse propensity weighting (IPW) to PU learning, which assigns a weight to each positive sample. Based on the weights, they obtained a new unbiased PU risk estimator. They also proposed a method that can compatible to deep networks to estimate the weights when the labeling mechanism is unknown.

**Strengths:**

* The problem is interesting and valuable. The adaptation of causal inference to PU learning is sound and original to the best of my knowledge.
* The resulting algorithm is unbiased, which is a very nice result (but I quickly checked the proofs and have some concerns).
* The algorithm is simple and easy to follow.
* The empirical studies well support the method's superiority with a clear margin.

**Weaknesses:**

* I'm a bit puzzled by the fact that the risk estimator is unbiased.

$R_{PN}$ in current version should be $\hat{R_{PN}}$ for notation consistency, while $R_{PN}$ denotes the expected risk. Then if $\mathbb{E}[\hat{R_{PUe}}(g)] = R_{PN}(g)$, we say $\hat{R_{PUe}}$ is unbiased.
While in this paper, the authors proved $\mathbb{E}[\hat{R_{PUe}}(g)] = \hat{R_{PN}}$ (Eq.11), which looks a little weird.

* The examples given by the authors give readers a very clear idea that biased selection is likely to be present in the PU problem, but it seems to me that Figure 1 doesn't present a very clear picture of the setting discussed in the paper. Wouldn't it be more telling if all the blue dots were centered, like, in the upper left corner? But if so could ePU also have difficulty learning good classification boundaries, since this bias could also seriously mislead the estimation of $e$ (step 1 in Figure 2)?
* $s$ should be introduced in Eq.4.
* In line 113, "it does not depend on...", "it" here seems very vague.
* Tables 1 and 2 are too dense a presentation of the results, and I would suggest labeling the best results more obviously.

**Questions:**

Please see the main review.

---

> ### Author Rebuttal · Authors · 2023-08-10
>
> *Q1:* **$R_{PN}$ should be $\hat{R_{PN}}$. Eq.11 looks a little weird.**
>
> *A1:* Thank you for pointing out the issue. It is true that the expression forms here are not uniform. The empirical risk should be represented by $\hat{R_{PN}}(g|y)=\frac{1}{n}\sum_{i=1}^{n}[y_iL(g(x_i),+1)+(1-y_i)L(g(x_i),-1)]$ and the expected risk should be represented by $R_{PN}(g|y)=E_{P(x,y)}(L(g(x),y))=E_i(\hat{R_{PN}}(g|y))$. Obviously, the expectation of empirical risk equals the expectation of expected risk. Using the two-stage expectation $\mathbb{E}_i[\mathbb{E}_s]$, we can obtain that the risk estimation of our algorithm is unbiased.
>
> $\mathbb{E}[\mathbb{E}(\hat{R_{PU_e}}(g))]=R_{PN}(g|y)$. It does not affect the conclusion. We'll add this to the final version of the paper.
>
> *Q2:* **If all the blue dots were centered, could PUe learn good classification boundaries? Or is it possible that the bias can mislead the estimation of $e$ (Step 1 in Figure 2)?**
>
> *A2:* Thank you for your question. The scenario you said is different from the condition setting of our paper and contradicts the Probabilistic Assignment assumption in causal inference. If all the blue dots were centered, like, in the upper left corner. Without additional assumptions, machine learning cannot learn good classification boundaries, and unlabeled positive classes cannot be correctly classified. Our scenario actually requires that all orthodox classes be labeled with probability, even if they are small, which means that Probabilistic Assignment is $0<P(s,x|y_i=1)<1$ for all samples.
>
> The scenario you're hypothetical, which is denoted as partial positive, has a tendency score of 0. For example, in the odd and even category, a scenario that doesn't select 8 at all for labeling is similar to your assumption. According to the no free lunch principle, the same algorithm does not exist for the same data, and the classification of 8 must be accurate. If algorithm A correctly classifies eight of [0, 2, 4, 6, 8] and [1, 3, 5, 7, 9], algorithm A can not correctly classifies eight of [0, 2, 4, 6] and [1, 3, 5, 7, 8, 9]. 8 is not labeled, so it is impossible to determine whether 8 is positive or negative without additional assumptions. In our scenario, even if the probability of 8 being labeled is very small, the accuracy of the final algorithm is improved as long as there is labeling. The results are as shown in the paper.
>
> *Q3:* **$s$ should be introduced in Eq.4.**
>
> *A3:* Sorry for the unclear statement. $s=1$ here means that the sample was selected into the set of annotations. $s=0$ indicates that the sample is not selected into the annotation set. We'll clearly state this symbol in the final version of the paper.
>
> *Q4:* **In line 113, "it does not depend on...", "it" here seems very vague.**
>
> *A4:* Thank you for your correction. Here "It" means the labeling mechanism.It's not clear here. We will refine it in the final version of the paper.
>
> *Q5:* **Tables 1 and 2 are too dense a presentation of the results, and I would suggest labeling the best results more obviously.**
>
> *A5:* Thanks for your advice. We will mark the best results in the final version of the paper.

---

> > ### Comment · Reviewer_U7Dp · 2023-08-17
> >
> > Thank you for your response. You have addressed all my concerns.

---

### Comment · Area_Chair_AZS8 · 2023-08-18
**Reviewer-Author Discussion Period**

Dear All,

Thank you reviewers for your hard work in evaluating this submission, and thank you authors for responding to the reviewers’ questions and concerns.

We are now entering the final phase of the discussion period, which will run until 21 Aug, and some of the authors' responses have to been acknowledged by all reviewers.

@Reviewers, if you have any follow up questions or comments on the rebuttal or the responses, now is the time to express them. At the very least, please acknowledge that you have read the authors’ response to your review.

Thank you everyone for making the review process a fruitful, constructive, and civil process.

AC

---

### Decision · Program_Chairs · 2023-09-21

**Decision:**

Accept (poster)

**Comment:**

This paper notes that existing Positive-Unlabeled (PU) learning methods assume that positive samples are selected entirely at random, ignoring the prevalent selection bias in real-world PU problems. To overcome this limitation, the authors propose a PU learning enhancement (PUe) algorithm based on causal inference theory. They adapt sample re-weighting methods, to be specific, inverse propensity weighting (IPW) to PU learning, which assigns a weight to each positive sample. Based on the weights, they obtain a new unbiased PU risk estimator. They also propose a method that can compatible to deep networks to estimate the weights when the labeling mechanism is unknown.

The strength of this paper includes the importance of the studied problem (biased positive data), writing and experiments. For example, the problem of studying biased positive data is highly significant and practical. In many real-world scenarios, the selection of positive samples is often biased, while existing methods typically assume that the selection of positive samples is completely random. The proposed PUe algorithm in this paper addresses this issue and effectively handles biased labeled data, filling a research gap in the field of PU learning. Moreover, this paper is well-written, with clear descriptions of the problem background, method principles, and experimental design and results. The authors introduce causal inference theory and utilize techniques such as normalized propensity scores and inverse probability weighting to provide a viable solution, which is presented in a clear and concise manner. Last but not least, experimental results demonstrate the effectiveness of the method: The paper provides experimental evidence of the proposed PUe algorithm's effectiveness.

Overall, the clarity and novelty are above the bar of NeurIPS. While the reviewers had some concerns, the authors did a particularly good job in their rebuttal. Thus, most of us have agreed to accept this paper for publication! Please include the additional experimental results in the next version.